# Efficient Personalized Federated Learning via Sparse Model-Adaptation

## Abstract

Federated Learning (FL) aims to train machine learning models for multiple clients without sharing their own private data. Due to the heterogeneity of clients' local data distribution, recent studies explore the personalized FL that learns and deploys distinct local models with the help of auxiliary global models. However, the clients can be heterogeneous in terms of not only local data distribution, but also their computation and communication resources. The capacity and efficiency of personalized models are restricted by the lowest-resource clients, leading to sub-optimal performance and limited practicality of personalized FL. To overcome these challenges, we propose a communication and computation efficient approach named pFedGate by adaptively and efficiently learning sparse local models. With a lightweight trainable gating layer, pFedGate enables clients to reach their full potential in model capacity by generating different sparse models accounting for both the heterogeneous data distributions and resource constraints. Meanwhile, the computation and communication efficiency are both improved thanks to the adaptability between the model sparsity and clients' resources. Further, we theoretically show that the proposed pFedGate has superior complexity with guaranteed convergence and generalization error. Extensive experiments show that pFedGate achieves superior global accuracy, individual accuracy and efficiency simultaneously over state-of-the-art methods. We also demonstrate that pFedGate performs better than competitors in the novel clients participation and partial clients participation scenarios, and can learn meaningful sparse local models adapted to different data distributions.

## 1 Introduction

Federated Learning (FL) gains increasing popularity in machine learning scenarios where the data are distributed in different places and can not be transmitted due to privacy concerns (Muhammad et al., 2020; Meng et al., 2021; Yu et al., 2021; Hong et al., 2021; Yang et al., 2021). Typical FL trains a unique global model from multiple data owners (clients) by transmitting and aggregating intermediate information with the help of a centralized server (McMahan et al., 2017; Kairouz et al., 2021). Although using a shared global model for all clients shows promising average performance, the inherent statistical heterogeneity among clients challenges the existence and convergence of the global model (Sattler et al., 2020; Li et al., 2020). Recently, there are emerging efforts that introduce personalization into FL by learning and deploying distinct local models (Yang et al., 2019; Karimireddy et al., 2020; Tan et al., 2021). The distinct models are designed particularly to fit the heterogeneous local data distribution via techniques taking care of relationships between the global model and personalized local models, such as multi-task learning (Collins et al., 2021), meta-learning (Dinh et al., 2020a), model mixture (Li et al., 2021c), knowledge distillation (Zhu et al., 2021) and clustering (Ghosh et al., 2020).

However, the heterogeneity among clients exists not only in local data distribution, but also in their computation and communication resources (Chai et al., 2019; 2020). The lowest-resource clients restrict the capacity and efficiency of the personalized models due to the following reasons: (1) The adopted model architecture of all clients is usually assumed to be the same for aggregation compatibility and (2) The communication bandwidth and participation frequency of clients usually determine how much can they contribute to the model training of other clients and how fast can they agree to meet a converged "central point" w.r.t their local models. This resource heterogeneity is

under-explored in most existing personalized FL (pFL) works, and instead, they gain the accuracy improvement with a large amount of additional computation or communication costs. Without special design taking the efficiency and resource heterogeneity into account, we can only gain sub-optimal performance and limited practicality of pFL.

To overcome these challenges, in this paper, we propose a novel method named pFedGate for efficient pFL, which learns to generate personalized sparse models based on adaptive gated weights and different clients' resource constraints. Specifically, we introduce a lightweight trainable gating layer for each client, which predicts sparse, continues and block-wise gated weights and transforms the global model shared across all clients into a personalized sparse one. The gated weights prediction is conditioned on specific samples for better estimation of the heterogeneous data distributions. Thanks to the adaptability between the model sparsity and clients' resource, the personalized models and FL training process gain better computation and communication efficiency. As a result, the model-adaption via sparse gated weights delivers the double benefit of personalization and efficiency: (1) The sparse model-adaption enables each client to reach its full potential in model capacity with no need for compatibility across other low-resource clients, and to deal with a small and focused hypothesis space that is restricted by the personalized sparsity and the local data distribution. (2) Different resource restrictions can be easily imposed on the predicted weights as we consider the block-wise masking under a flexible combinatorial optimization setting. We further provide space-time complexity analysis to show pFedGate' superiority over state-of-the-art (SOTA) methods, and provide theoretical guarantees for pFedGate in terms of its generalization and convergence.

We evaluate the proposed pFedGate on four FL benchmarks compared to several SOTA methods. We show that pFedGate achieves superior global accuracy, individual accuracy and efficiency simultaneously (up to 4.53% average accuracy improvement with 12x smaller sparsity than the compared strongest pFL method). We demonstrate the effectiveness and robustness of pFedGate in the partial clients participation and novel clients participation scenarios. We also find that pFedGate can learn meaningful sparse local models adapted to different data distributions, and conduct various experiments to verify the necessity and effectiveness of pFedGate' components.

Our main contributions can be summarized as follows:

- We exploit the potential of co-design of model compression and personalization in FL, and propose a novel computation and communication efficient pFL approach that learns to generate sparse local models with a fine-grind sample-level adaptation.

- We provide a new formulation for the efficient pFL considering the clients' heterogeneity in both local data distribution and hardware resources, and provide theoretical results about the generalization, convergence and complexity of the proposed method.

- We achieve SOTA results on several FL benchmarks and illustrate the feasibility of gaining better efficiency and accuracy for pFL simultaneously. To facilitate further studies, we release our code at https://github.com/AnonyMLResearcher/pFedGate.

## 2 RELATED WORKS

**Personalized FL.** Personalized FL draws increasing attention as it is a natural way to improve FL performance for heterogeneous clients. Many efforts have been devoted via multi-task learning (Smith et al., 2017; Corinzia & Buhmann, 2019; Huang et al., 2021; Marfoq et al., 2021), model mixture (Zhang et al., 2020; Li et al., 2021c), clustering (Briggs et al., 2020; Sattler et al., 2020; Chai et al., 2020), knowledge distillation (Lin et al., 2020; Zhu et al., 2021; Ozkara et al., 2021), meta-learning (Khodak et al., 2019; Jiang et al., 2019; Khodak et al., 2019; Fallah et al., 2020; Singhal et al., 2021), and transfer learning (Yang et al., 2020; He et al., 2020; Zhang et al., 2021a). Although effective in accuracy improvements, most works pay the cost of additional computation or communication compared to non-personalized methods. For example, Sattler et al. (2020) consider group-wise client relationships, requiring client-wise distance calculation that is computationally intensive in cross-device scenarios. Fallah et al. (2020) leverage model agnostic meta learning to enable fast local personalized training that requires computationally expensive second-order gradients. Zhang et al. (2021a) learn pair-wise client relationships and need to store and compute similarity matrix with square complexity w.r.t. the number of clients. Marfoq et al. (2021) learn a mixture of multiple global models which multiplies the storing and communicating costs. Our work differs from these works by considering a practical setting that clients are heterogeneous in both the data distribution and hardware resources. Under this setting, we achieve personalization from a novel

perspective, personalized sparse model-adaptation, and show the feasibility of gaining better accuracy, and computation and communication efficiency at the same time.

**Efficient FL.** Fruitful FL literatures have explored the improvement of communication efficiency such as methods based on gradient compression (Rothchild et al., 2020; Alistarh et al., 2017; Reisizadeh et al., 2020; Haddadpour et al., 2021; Zhang et al., 2021b), model ensemble (Hamer et al., 2020), model sub-parameter sharing (Liang et al., 2020) sub-model selection Tianchun et al. (2022); Minh & Carl (2021), and Bayesian neural network (Yurochkin et al., 2019b;a). Few works improve computation and communication efficiency with personalized local models via quantization (Ozkara et al., 2021; Li et al., 2021a; Hong et al., 2022) and model parameter decoupling (Diao et al., 2021; Collins et al., 2021; Huang et al., 2022). The most important difference is that our method differs from these works by adaptively generating the sparse model weights at a fine-grained sample level, which achieves a good estimation for the conditional probability of heterogeneous local data and high accuracy as shown in our experiments. Due to the space limit, we present more detailed description and comparison with some of these works in Appendix E

## 3  PROBLEM FORMULATION

The goal of traditional federated learning (FL) is to fit a single global model by collaboratively training models from a set $\mathcal{C}$ of clients without sharing their local data. In this paper, we focus on the personalized federated learning problem in terms of not only client-distinct models, but also client-distinct resource limitations. Specifically, consider each client $i \in \mathcal{C}$ has its own private dataset $\mathcal{S}_i$ that is drawn from a local distribution $\mathcal{D}_i$ over $\mathcal{X} \times \mathcal{Y}$. In general, the local data distributions $\{\mathcal{D}_i\}_{i \in \mathcal{C}}$ are heterogeneous and thus it is promising to learn personalized model $h_{\theta_i} \in \mathcal{H} : \mathcal{X} \mapsto \mathcal{Y}$ parameterized by $\theta_i$ for each local distribution $\mathcal{D}_i$, where $\mathcal{H}$ is the set of hypotheses with $d$ dimensions. Besides, clients may have heterogeneous computation and communication resources especially on cross-device FL scenarios. To better account for the data heterogeneity and system heterogeneity, we aim to optimize the following objective:

$$\min_{\{h_{\theta_i}\}_{i \in \mathcal{C}}} \sum_{i \in \mathcal{C}} p_i \cdot \mathbb{E}_{(x,y) \sim \mathcal{D}_i}[f(\theta_i; x, y)],$$
$$s.t. \quad size(\theta_i) \leq d_i, \quad \forall i \in \mathcal{C}, \tag{1}$$

where $f(\theta_i; x, y) \triangleq \ell(h_{\theta_i}(x), y)$ and $\ell : \mathcal{Y} \times \mathcal{Y} \mapsto \mathbb{R}^+$ is the loss function, $p_i$ is non-negative aggregation weight for client $i$ and $\sum_{i \in \mathcal{C}} p_i = 1$. In typical FL setting (McMahan et al., 2017; Kairouz et al., 2021), $p_i$ indicates the participation degree of client $i$ and is set to be proportional to the size of local dataset $|\mathcal{S}_i|$. The $size(\theta_i)$ indicates the number of parameters of $\theta_i$, and $d_i$ indicates the model size limitation for client $i$. Without special designs to handle the system heterogeneity (here the size restriction), most pFL methods simply adopt another hypothesis space $\tilde{\mathcal{H}}$ with $\min(\{d_i\}_{i \in \mathcal{C}})$ dimensions that are constraints by the lowest-resource clients, and the problem is degraded to the typical one that minimizes the objective without constraints. We consider the resource constraints in terms of model size limitation in this paper as it fairly reflect the computation and communication cost. Later we will see that our approach can be extended to other configurations such as FLOPs (Floating-point operations per second), latency and communication bandwidth.

## 4  LEARNING TO GENERATE SPARSE MODEL

In practice, one can solve the problem in Equation (1) with a two-step manner: ① find optimal local models via existing personalized federated learning methods without sparsity constraints, and then ② compress the trained models into required local model size via compression techniques such as pruning, quantization and knowledge distillation (Deng et al., 2020). However, the final local models from step ② usually gain worse performance to models found in step ① (Jiang et al., 2022). Besides, the post-compression process still requires computational and communication costs corresponding to the un-compressed models during the FL process. To alleviate the performance degradation and further improve the efficiency of FL process, we propose to jointly learn and compress the federated models. Specifically, we can directly learn sparse local models whose number of non-zero parameters satisfy the size requirements:

$$\min_{\{h_{\theta_i'}\}_{i \in \mathcal{C}}} \sum_{i \in \mathcal{C}} p_i \cdot \mathbb{E}_{(x,y) \sim \mathcal{D}_i}[\ell(h_{\theta_i'}(x), y)], \tag{2}$$

where $h_{\theta_i'} \in \mathcal{H}_{d_i}$ and $\mathcal{H}_{d_i}$ indicates the subset of $\mathcal{H}$ with hypotheses whose number of non-zero parameters is not larger than $d_i$. However, the sparse models $h_{\theta_i'}$ can be arbitrarily different across clients due to the various dimension requirements $d_i$ and local data distributions $\mathcal{D}_i$. Directly optimizing $h_{\theta_i'}$ and aggregating them may lead to sub-optimal performance and make the federated

learning process un-convergent. As Marfoq et al. (2021) discussed, clients can not benefit from each other without any additional assumption to the local distributions. Here we consider the case of bounded diversity as following assumption shows:

**Assumption 1.** *(Bounded Diversity) Let $\mathcal{D}_g$ be a global data distribution that is a mixture of all local data distribution $\mathcal{D}_i$, $i \in \mathcal{C}$. Denote $h_{\theta_g^*}$ and $h_{\theta_i^*}$ as the optimal estimator for the conditional probability of $\mathcal{D}_g$ and $\mathcal{D}_i$ respectively. There exist a $\mathcal{D}_g$ such that the variance between local and global gradients is bounded*

$$||\nabla_\theta f(\theta_i^*) - \nabla_\theta f(\theta_g^*)||^2 \leq \sigma_i^2, \quad \forall i \in \mathcal{C}, \tag{3}$$

*where $f(\theta)$ is a compact notation as $f(\theta) \triangleq f(\theta; x, y)$ for all data points $(x, y) \in \mathcal{X} \times \mathcal{Y}$.*

**Proposition 1.** *Denote $(\theta_i')^*$ be the parameter of any sparse optimal estimator for $\mathcal{D}_i, i \in \mathcal{C}$. If $f$ is $\mu$-strongly convex, under Assumption 1, there exist continuous weights $M_i^* \in \mathbb{R}^d$ such that $(\theta_i')^* = \theta_g^* \odot M_i^*$ where $\odot$ indicates element-wise production over model parameters of $\theta$, and $\{M_i^*\}_{i \in \mathcal{C}}$ can be bounded as*

$$||(M_i^* - 1)|| \leq \frac{\sigma_i^2}{\mu^2 ||\theta_g^*||^2} \cdot \frac{R_{\theta_i} R_{\theta_g}}{r_{\theta_i} r_{\theta_g}} \quad \forall i \in \mathcal{C}, \tag{4}$$

*where $r_{\theta_i}$ and $r_{\theta_g}$ are the infimum of $\theta_i$ and $\theta_g$ respectively, $R_{\theta_i}$ and $R_{\theta_g}$ are the supremum of $\theta_i$ and $\theta_g$ respectively.*

The proofs of all propositions and theorems of our work are in Appendix. Proposition 1 suggests that we can deal with Problem (2) via indirectly generating personalized sparse models $h_{\theta_i'}$ parameterized by $\theta_i' = \theta_g \odot M_i$ with personalized weights $M_i$ and a shared global model $\theta_g$, which helps to transfer information cross clients and leads to a controllable optimization. Specifically, denote $s_i$ be the model sparsity constraint of client $i$, Problem (2) can be transformed into following form:

$$\min_{\theta_g, \{M_i\}_{i \in \mathcal{C}}} \sum_{i \in \mathcal{C}} p_i \cdot \mathbb{E}_{(x,y) \sim \mathcal{D}_i}[f(\theta_g \odot M_i; x, y)],$$
$$s.t. \quad count(M_i \neq 0)/d \leq s_i, \quad \forall i \in \mathcal{C}. \tag{5}$$

With this re-formulation, $M_i$ can be regarded as sparsity-enforced gated weights that absorb the diversity of different local data distribution into the global model, via *scaling* and *blocking* the information flow through the shared model $\theta_g$.

## 5 PERSONALIZED FL WITH CONDITIONAL GATED WEIGHTS

### 5.1 OVERALL FRAMEWORK

Note that for discriminative models (*e.g.*, neural networks), the personalized models are estimators for conditional distribution $P_{\mathcal{D}_i}(y|x)$ associated to clients' local data $\mathcal{D}_i$. To achieve efficient pFL and optimize the objective described in Equation (5), instead of learning client-level element-wise weights $M_i \in \mathbb{R}^d$, we propose to learn to predict sample-level block-wise weights $M_{i,j}' = g_{\phi_i}(x_{i,j})$, where $\phi_i$ indicates the parameters of a trainable personalized gating layer, and $g_{\phi_i}$ is the predictive function conditioned on specific data sample $x_{i,j}$ and sparsity limitation $s_i$. The gating layer $\phi_i$ brings up benefits in both efficiency and effectiveness: (1) predicting block-wise weights $M_{i,j}' \in \mathbb{R}^L$ enables us to use a lightweight $\phi_i$ for $L \in \mathbb{N}^+$ sub-blocks of $\theta$ with $L \ll d$, and thus gain much smaller computational costs than the element-wise manner; (2) predicting sample-wise weights can estimate the conditional probability better than the client-level manner since each client may have complex local data distribution mixed by different classes, *e.g.*, using different sparse models to classify dog and bird can achieve better performance than using the same one. We empirically verify the effectiveness of these choices in Section 7.4.

Based on the introduced gating layer, we establish an efficient pFL framework in server-client setting as shown in Figure 1. To enable benefits from each other, all clients share the same global model $\theta_g$ that downloaded from server. Each client $i \in \mathcal{C}$ has a private, lightweight and learnable gating layer $\phi_i$, which is trained from local data with a differentiable manner, and learns to achieve fine-grained personalized model-adaptation $\theta_{i,j}' = \theta_g \circledast M_{i,j}' = \theta_g \circledast g_{\phi_i}(x_{i,j})$ given different data samples, where $\circledast$ indicates block-wise production of $\theta$. To handle diverse resource-limited scenarios, $\phi_i$ ensures that the personalized adapted model $\theta_{i,j}'$ has required sparsity not larger than $s_i$. Moreover, the sparsity speeds up the training and inference of local models, as well as the communication via uploading the sparse model updates (we give detailed discussion on Sec.6.3 and Appx. E).

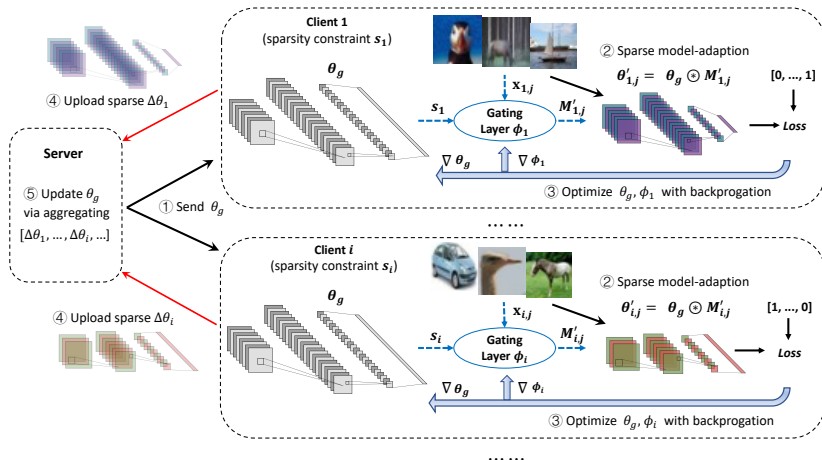

Figure 1: The pFedGate framework that learns to generate sparse personalized models conditioned on clients' different data samples and resource limitations. The serial numbers indicate the FL processes at each communication round.

## 5.2 ADAPTIVE GATING LAYER

### 5.2.1 LAYER ARCHITECTURE

We adopt a simple two-path sandwich architecture for the gating layer as shown in the left part of Figure 2. One path predicts the block-wise gated weights $M$ that may violate the hard sparsity constrains, and the other path predicts the block-wise importance scores that are used to adjust $M$ into a sparse one $M'$ such that the final sparse model satisfies the hard limitation (we introduce the adjustment later). Let the number of sub-blocks of $\theta_g$ be $L$ and the dimension of flatten input feature be $d_X$, *i.e.*, $x \in \mathbb{R}^{d_X}$. The middle two fully connected layers have the same size and each of them has $(d_X \cdot L)$ parameters. Here the switchable normalization (Luo et al., 2019) is used to handle batched data samples and different input

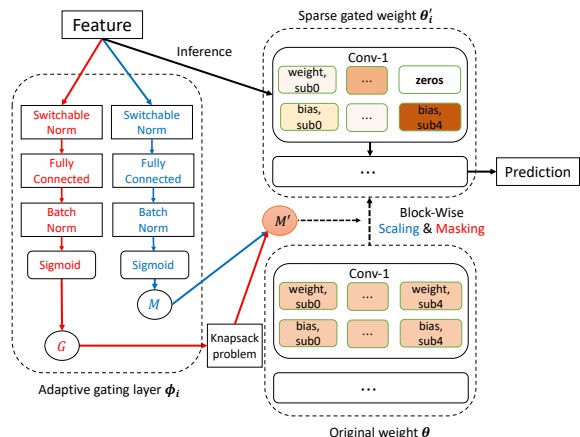

Figure 2: The sparse model-adaptation via gating layer.

types via learning mixture of batch norm, instance norm and layer norm with negligible additional number of trainable parameters. Furthermore, the personalized normalization parameters within client-distinct gating layers enable our model-adaptation to absorb different feature distributions (similar empirical evidence is shown by Li et al. (2021d)). The final batch normalization and sigmoid activation are used to stabilize the optimization of $\theta_g$ and $\{\phi_i\}$ by bounding $M'$ and thus the scaling degrees on $\theta_g$'s parameters (recall that the upper bound in Proposition 1 is dependent on the infimum and supremum of parameters after scaling). In total, the gating layer learns and predicts the sparse gated weights $M' = g_\phi(x) = g(\phi; x)$ given specific data sample $x$ where $g : \mathbb{R}^{d_X} \mapsto \mathbb{R}^L$, with a lightweight parameter size $d_\phi = 2d_X L$ that is usually much smaller than the size of model $\theta_g$. We show the effectiveness and necessity of these adopted components of gating layer in Section 7.4.

### 5.2.2 OPERATOR-TYPE-FREE BLOCK SPLITTING

There are many alternatives to split a model into certain sub-blocks, *e.g.*, a convolutional operator can be split in channel-wise or filter-wise and an attention operator can be split in head-wise. Our gating layer is decoupled with specific block splitting manners and thus can easily support users' various splitting preferences. Besides, we propose a general operator-type-free splitting manner by default to enhance the usability of our approach as illustrated in the right part of Figure 2. Denote $B \in \mathbb{N}^+$ as a given block split factor, and denote $d_l$ as the dimension of a flatten vector that parameterizes the given learnable operator. We group the vector's first $\lfloor d_l \cdot s_{min} \rfloor$ elements as the first sub-block, and equally split the remaining elements into $(B-1)$ sub-blocks (the last sub-block may has fewer

size than $\lfloor (d_l \cdot (1 - s_{min})/(B - 1) \rfloor)$. Here $B$ provides flexible splitting granularity, $s_{min} \in (0, s_i]$ indicates a pre-defined minimal sparsity factor and the predicted gated weights for the first sub-block is enforced to be non-zero to avoid cutting information flow through this operator. The proposed splitting is simple to implement but effective for pFL as shown in our empirical evaluations.

### 5.2.3 SPARSE GATED WEIGHTS

To ensure an equal or smaller sparsity of adapted model than the limitation $s_i$, we propose to leverage the predicted block importance $G$ to adjust the predicted gated weights $M$ into a sparse one $M'$. Specifically, denote $W \in \mathbb{Z}^L$ be the parameter size look-up table of $L$ sub-blocks, we transform $M' = M \cdot I^*$ where $I^* \in \{0, 1\}^L$ is binary index and a solution of the following knapsack problem that maximizes total block importance values while satisfy the sparsity constraint:

$$\max_I \quad I \cdot G, \qquad s.t. \quad |I \cdot W|/d \leq s_i. \tag{6}$$

In practical, to enable gradient-based optimization of gating layer, we leverage the straight-through trick (Hubara et al., 2016) via differentiable $G$ in backward pass. In Appendix A, we summarize the overall algorithm and give detailed descriptions about the gradients flows through the gating layer.

## 6 ALGORITHM ANALYSIS

### 6.1 GENERALIZATION

Following the parameter-sharing analysis from Baxter (2000) and the generalization analysis from Shamsian et al. (2021), here we give the generalization guarantee of our sparse model.

Based on the notations used in previous sections, let $\hat{\mathcal{L}}(\theta_g, \phi_i)$ denote the empirical loss of the sparse model in client $i$, and $\hat{\mathcal{L}}(\theta_g, \phi_i) \triangleq \frac{1}{n} \sum_{i=1}^n \sum_{(x,y) \in \mathcal{S}_i} f\left( (\theta_g \circledast g_{\phi_i}(x)); x, y \right)$. The expected loss is denoted as $\mathcal{L}(\theta_g, \phi_i) \triangleq \frac{1}{n} \sum_{i=1}^n \mathbb{E}_{(x,y) \sim \mathcal{D}_i} f\left( (\theta_g \circledast g_{\phi_i}(x)); x, y \right)$.

**Assumption 2.** *The parameters of global model and the gating layer can be bounded in a ball with radius $R$. The following Lipschitz conditions hold: $|f(\theta; x, y) - f(\theta'; x, y)| \leq L_f ||\theta_g - \theta'||$, $||h(\theta_i; x) - h(\theta_i'; x)|| \leq L_h ||\theta_i - \theta_i'||$ and $||g(\phi_i; x) - g(\phi_i'; x)|| \leq L_g ||\phi_i - \phi_i'||$, where $h(\theta_i; \cdot)$ is the sparse model parameterized by $\theta_i$ and $g(\phi_i; \cdot)$ indicates the gating layer parameterized by $\phi_i$.*

Let the parameter spaces of the sparse model and gating layer be of size $d$ and $d_\phi$, separately. With the above assumption, we have the following theorem regarding the generalization error bound.

**Theorem 1.** *Under Assumption 2, there exist $\tilde{N} =$*

$$\mathcal{O}\left( \frac{d}{|\mathcal{C}|\epsilon^2} \log \frac{RL_f L_h (RL_g + 1)}{\epsilon} + \frac{d_\phi}{\epsilon^2} \log \frac{RL_f L_h (RL_g + 1)}{\epsilon} - \frac{\log \delta}{|\mathcal{C}|\epsilon^2} \right)$$

*such that for all $\theta_g, \phi_i$, with probability at least $1 - \delta$, $|\mathcal{L}(\theta_g, \phi_i) - \hat{\mathcal{L}}(\theta_g, \phi_i)| \leq \epsilon$ when the number of client $i$'s local data samples is greater than $\tilde{N}$.*

Theorem 1 indicates the generalization depends on the size of the global model (*i.e.*, $d$) reduced by the number of clients $|\mathcal{C}|$ (the first term), as the global model is shared by all clients. It also depends on the size of the gating layers (the second term), and it is not reduced by $|\mathcal{C}|$ since each client has its own personalized parameters. Besides, the generalization is also affected by the Lipschitz constants of the global model, sparse model, and the gating layer, as they together constrain the parameter space that our method can search.

### 6.2 CONVERGENCE

Under the following a few mild assumptions that are commonly used in convergence analysis of many FL works (Kairouz et al., 2021; Li et al., 2020) (more discussion about these assumptions are in Appendix D), we can see that the learnable parameters of pFedGate ($\theta_g$ and $\{\phi_i\}_{i \in \mathcal{C}}$) converge to stationary points, meanwhile the global model updates $\{\Delta \theta_{g,i}^t\}_{i \in \mathcal{C}_s}$ become arbitrarily small.

**Assumption 3.** *(Smoothness) For all clients $i \in \mathcal{C}$, $(x, y) \in \mathcal{D}_i$ and all possible $\theta_g$, the function $f(\theta_g; x, y) \triangleq \ell(h_{\theta_g}(x), y)$ is twice continuously differentiable and $L$-smooth: $||\nabla f(\theta_g'; x, y) - \nabla f(\theta_g; x, y)||^2 \leq L||\theta_g' - \theta_g||$.*

**Assumption 4.** *(Bounded Output) The function $f$ is bounded below by $f^* \in \mathbb{R}$.*

**Assumption 5.** *(Bounded variance) For client $i \in \mathcal{C}$ and all possibile $\theta_g$, given data sample $(x, y)$ drawn from local data $\mathcal{S}_i$, the local gradient estimator $\nabla_\theta \ell(h_{\theta_g}(x), y)$ is unbiased and has bounded variance $\sigma^2$.*

**Theorem 2.** *Under Assumptions 1–5, if clients use SGD as local optimizer with learning rate $\eta$, there exist a large enough number of communication rounds $T'$, such that pFedGate converges with $\eta = \frac{\eta_0}{\sqrt{T'}}$:*

$$\frac{1}{T'} \sum_{t=1}^{T'} \mathbb{E} \left\| \nabla_\theta f\left( (\theta_g^t \circledast g_{\phi_i^t}(x)); x, y \right) \right\|^2 \leq \mathcal{O}\left( \frac{1}{\sqrt{T'}} \right), \ \forall i \in \mathcal{C}, \tag{7}$$

*where the expectation is over the random samples and $\mathcal{O}(\cdot)$ hides polylogarithmic factors.*

### 6.3 SPACE-TIME COMPLEXITY

We briefly summarize the efficiency of the proposed pFedGate method in terms of space-time costs. The training and inference cost of pFedGate is $\mathcal{O}\left(d(s_i + s_{\phi_i})\right)$ for client $i$ at each round, where the model sparsity $s_i$ and relative sparsity of gating layer $s_{\phi_i} = cound(\phi_i \neq 0)/d$ are usually small and lead to $(s_i + s_{\phi_i}) < 1$. By contrast, the computation cost of FedAvg and several SOTA pFL methods is at least $\mathcal{O}(d)$. The introduced sparsity provides great potential for boosting computation efficiency. As for the communication, the upload parameter number of pFedGate is $\mathcal{O}(q_i d)$ that can be also smaller than the one of baselines, $\mathcal{O}(d)$. Here the $q_i = count(\Delta\theta_i \neq 0)/d$ indicates the ratio of non-zero model updates, which depends on $s_i$ and local samples trained in the round. In Appendix E, we provide detailed discussions to show superiority of pFedGate over several SOTA competitors.

## 7 EXPERIMENTS

### 7.1 EXPERIMENTAL SETTINGS

We adopt four widely used FL datasets in our experiments: EMNIST (Cohen et al., 2017), FEMNIST (Caldas et al., 2018), CIFAR10 and CIFAR100 (Krizhevsky, 2009). They are partitioned into several sub-datasets to simulate the local dataset for a client, and each of them is randomly split into train/val/test datasets with ratio 6:2:2. In the experiments, the data partition follows the heterogeneous settings adopted by Marfoq et al. (2021); Dinh et al. (2020b). We evaluate pFedGate in terms of both personalization and FL efficiency. For personalization, we use standard FedAvg (McMahan et al., 2017) and FedAvg with fine-tuning (*FedAvg-FT*) as the baselines, and also choose several SOTA pFL methods, including *pFedMe* (Dinh et al., 2020b), *LG-FedAvg* (Liang et al., 2020), *Ditto* (Li et al., 2021c) and *FedEM* (Marfoq et al., 2021). As for efficiency, we compare several SOTA pFL methods that improve efficiency via shared model representations (*FedRep* (Collins et al., 2021)), binary quantization (*FedMask* Li et al. (2021a)), and sparse sub-models (*HeteroFL* (Diao et al., 2021)). Due to limited space, please refer to Appendix G for more details about the datasets, models and baselines.

### 7.2 OVERALL PERFORMANCE

**Clients' Average Performance.** We first examine the overall performance by evaluating the accuracy on local test set of each client, and averaging the results of all clients with weights proportional to their local dataset sizes. The detailed overall performance is shown in Table 1, where $\bar{s'}$ indicates the average of finally achieved sparsity across all clients (our block-wise operation may lead to $\bar{s'} \leq s$). We can see that pFedGate achieves better accuracy (averaged 3.1% to 4.53% accuracy improvements) and smaller sparsity even with $s$=0.3 at the same time. Compared with FedEM, the baseline with best accuracy, pFedGate doesn't have the heavy storage burden, since FedEM needs to store multiple (3) global models in the client. These observation demonstrate the effectiveness and efficiency of the proposed method. Note that compared to the SOTA efficient method HeteroFL, our method achieves better accuracy while with smaller sparsity (0.25 v.s. 0.55). When $s$=1, we achieve the best performance without sparsity constraints, verifying that the sample-level block-wise adaptation has a strong ability to achieve personalization. When $s$<1, we still gain comparable performance with $s$=1, providing evidence that there exist well-performed sparse models and they can be effectively learned by our approach.

Table 1: Accuracy comparison on widely-adopted FL datasets. $\overline{Acc}$ (%) and $\breve{Acc}$ (%) indicates the average accuracy of all clients and accuracy of the bottom decile clients respectively. $\bar{s}'$ indicates the average of finally achieved sparsity across all clients. Bold and underlined numbers indicate the best and second-best results respectively.

| | EMNIST | | FEMNIST | | CIFAR10 | | CIFAR100 | | Average | |
| | $\overline{Acc}$ | $\breve{Acc}$ | $\overline{Acc}$ | $\breve{Acc}$ | $\overline{Acc}$ | $\breve{Acc}$ | $\overline{Acc}$ | $\breve{Acc}$ | $\overline{Acc}$ | $\breve{Acc}$ |
|---|---|---|---|---|---|---|---|---|---|---|
| Local | 71.91 | 64.28 | 71.12 | 57.93 | 61.91 | 53.71 | 26.95 | 21.13 | 57.97 | 49.26 |
| FedAvg | 82.54 | 75.07 | 76.51 | 60.82 | 68.33 | 60.22 | 35.21 | 29.82 | 65.65 | 56.48 |
| FedAvg-FT | 83.19 | 76.83 | 78.43 | 64.22 | 69.91 | 61.55 | 37.15 | 30.91 | 67.17 | 58.38 |
| pFedMe | 83.29 | 76.48 | 75.29 | 57.63 | 70.29 | 62.13 | 36.14 | 30.13 | 66.25 | 56.59 |
| LG-FedAvg | 83.25 | 76.14 | 75.13 | 57.62 | 69.54 | 61.86 | 36.72 | 30.51 | 66.16 | 56.53 |
| Ditto | 82.75 | 78.12 | 79.29 | 63.24 | 73.14 | 62.59 | 37.32 | 30.11 | 68.13 | 58.52 |
| FedEM | 83.41 | 76.59 | 80.12 | 64.81 | 72.43 | 62.88 | 38.28 | 31.04 | 68.56 | 58.83 |
| FedRep | 82.42 | 77.71 | 79.17 | 62.89 | 72.77 | 63.44 | 36.68 | 30.51 | 67.76 | 58.64 |
| FedMask, $\bar{s}'$=0.5 | 81.95 | 77.26 | 78.69 | 62.18 | 72.43 | 62.88 | 36.21 | 30.13 | 67.32 | 58.11 |
| HeteroFL, $\bar{s}'$=0.55 | 85.64 | 77.76 | 77.22 | 59.13 | 70.97 | 63.64 | 37.01 | 31.27 | 67.71 | 57.95 |
| pFedGate, $s$=1 | 87.11 | 81.43 | **87.32** | **77.14** | **75.18** | **66.67** | **42.01** | **35.03** | **72.91** | **65.07** |
| $s$=0.5, $\bar{s}'$=0.43 | **87.28** | 81.15 | 86.31 | 75.68 | 74.07 | 64.21 | 40.07 | 32.38 | 71.93 | 63.36 |
| $s$=0.3, $\bar{s}'$=0.25 | 87.09 | **82.52** | 86.75 | 76.47 | 73.65 | 64.39 | 39.53 | 31.63 | 71.76 | 63.75 |

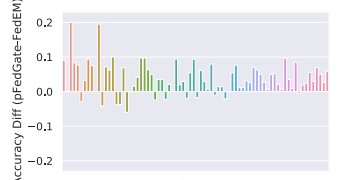

Figure 3: The client-wise accuracy difference between pFedGate and FedEM on CIFAR10 dataset.

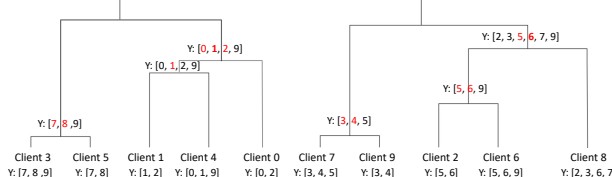

Figure 4: The hierarchical clustering result for parameters of clients' gating layers on CIFAR-10 with 10-clients partition.

**Individual Client Performance.** We then examine whether pFedGate improved the average performance by sacrificing the performance of some clients. In Table 1, we mark the accuracy of bottom decile local models $\breve{Acc}$ as the $\lfloor |\mathcal{C}|/10 \rfloor$-th worst accuracy (following Marfoq et al. (2021), we neglect the particularly noisy results from clients with worse accuracy due to their very small local data sizes). We find that pFedGate also gains significant $\breve{Acc}$ improvement (averaged 4.92% to 6.24% than FedEM). We plot the client-wise accuracy difference between our method with $s = 0.5$ and the strongest baseline, FedEM in Figure 3, in which pFedGate significantly improves most clients compared to the strongest baseline. The results in Table 1 and Figure 3 show that pFedGate not only improves the average accuracy and efficiency, but also fairly improves the individual client performance by learning to generate personalized models with adaptive gating layers.

### 7.3 Personalization Study

We propose to learn personalized gating layers $[\phi_i]$ that estimate the heterogeneous data distribution with sparse model-adaption. To investigate the level of personalization achieved by pFedGate, we conduct experiments on another CIFAR10 non-i.i.d. partition with 10 clients and pathological splitting (McMahan et al., 2017) that sorts the data by labels and assigns to clients with equal-sized shards.

We do hierarchical clustering for the parameters of learned gating layers $[\phi_i]$ with Ward's method (Ward Jr, 1963) and euclidean distance. We illustrate the results in Figure 4 when the models of clients converge with $s$=0.3 and achieve good performance ($\overline{Acc}$=89.21%), where we mark the local label categories of clients and common label categories after merging in red numbers. Interestingly, we can see that clients learn gating layers with smaller parameter distances along with they share more similarities in label categories. This indicates that pFedGate not only can achieve good client-distinct personalization, but also implicitly learn the group-wise personalization with a strong sparsity constraint, which could be beneficial to applications where there are inherent partitions among clients.

Table 2: Average accuracy and round to achieve $0.8 \cdot \overline{Acc}$ accuracy ($T_{0.8}$) normalized by standard pFedGate.

| | EMNIST | | CIFAR100 | |
| --- | --- | --- | --- | --- |
| | $\overline{Acc}$ | $T_{0.8}$ | $\overline{Acc}$ | $T_{0.8}$ |
| pFedGate, s=0.5 | 87.28 | 1 | 39.72 | 1 |
| w/o pre-norm | 86.25 | 1.13 | 38.32 | 1.24 |
| w/o post-norm | 87.14 | 1.10 | 39.67 | 1.43 |
| w/o sigmoid | 83.55 | 1.85 | 31.29 | 1.93 |
| client-wise adaptation | 84.83 | 0.74 | 33.52 | 0.86 |

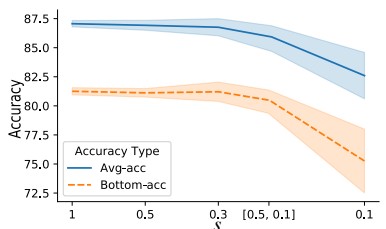

Figure 5: The accuracy on EMNIST when varying $s$.

### 7.4 ABLATION STUDY

**Choices of Gating Layer.** To gain further insight into our method, we ablate the gating layer w.r.t. its switchable norm layer (*pre-norm*) and batch norm layer (*post-norm*) within the sandwich structure and the *sigmoid* activation that bounds the gated weights. We also change the sample-wise adaption into *client-wise* manner by feeding the input feature mean-pooled over all local training data samples of each client. We show the results in Table 2 with $s$=0.5.

We find that the two normalization layers improve the convergence speed. Beside, *pre-norm* has a larger impact on performance than *post-norm* since it helps to handle the heterogeneous feature distribution. The *sigmoid* significantly impacts both the convergence and accuracy, showing effectiveness of gated weights bounding operation, which restricts the dissimilarity level between the personalized model and global model, and in turn helps to learn a global model suitable for different clients. Compared to sample-level manner, although *client-wise adaption* achieves a faster training, it pays cost of significant accuracy drop due to the coarser-grained modeling of conditional distribution of local data and smaller parameter space could be explored under sparsity constraints. In a nutshell, the ablation results verify effectiveness and necessity of our choices in gating layer.

**Effect of $s$.** We further study the effect of the sparsity factor $s$ by varying $s$ from 1 to 0.1 on EMNIST dataset and illustrate the results in Figure 5, in which "$[0.5, 0.1]$" indicates that we randomly divide all clients into two equal-sized parts and set $s$ to be 0.5 and 0.1 for these two parts respectively. We can see that our method is robust to keep a good performance when adopting not very small sparsity degrees. Another observation is that the performance variance increases as $s$ decreases. Besides, our method still performs well in a sparsity-mixed setting, verifying the robustness of our method again.

### 7.5 OVERVIEW OF MORE EXPERIMENTS IN APPENDIX

Due to the space limitation, we provide further experiments and analysis in Appendix in terms of

- Generalization: In Appx.H.1, we present additional results of pFedGate for the un-seen clients that haven't participated in the FL training stage (Diao et al., 2021; Marfoq et al., 2021; Yuan et al., 2022), in which case pFedGate achieves better performance than baselines since it only needs to train the personalized lightweight gating layers with a small number of parameters.

- Robustness: In Appx.H.2, we conduct experiments in partial clients participation case with sampling technique to further verify the effectiveness and robustness of pFedGate in practical FL scenarios.

- Convergence and model compression manner: In Appx.H.3, we show that pFedGate can achieve effective optimization, which supports the Theorem 2. In Appx.H.4, we vary the block splitting factor $B$ of pFedGate and make more detailed comparison to FedMask (Li et al., 2021a).

## 8 CONCLUSION

Existing personalized Federated Learning (pFL) methods usually focus on the heterogeneity of local data distribution only and pay additional computation and communication costs. In this paper, we explore the organic combination of model compression and pFL by adaptively generating different sparse local models at a fine-grained sample level, which gains double benefit of personalization performance and efficiency. Theoretical analysis are provided w.r.t. generalization, convergence and space-time complexity. Extensive experiments are conducted to verify the effectiveness, efficiency and robustness of the proposed method in various benchmarks and scenarios. We demonstrate the feasibility of obtaining superior accuracy and efficiency simultaneously for pFL, and believe this work can enlighten more studies on efficient and practical pFL.

## REPRODUCIBILITY STATEMENT

For the proposed proposition and theorems, we give the complete proof in Appendix B (for proposition 1), Appendix C (for theorem 1) and Appendix D (for theorem 2). More introduction and implementation details of experimental setups are also presented in Appendix G, including configurations for the datasets, models and baselines and our method. To facilitate the reproducibitlity, we summarize the algorithm of the proposed method in Appendix A and provide the source codes in the anonymous link, https://github.com/AnonyMLResearcher/pFedGate.

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

APPENDICES FOR THE SUBMISSION: EFFICIENT PERSONALIZED FEDERATED LEARNING VIA SPARSE MODEL-ADAPTATION

We provide more details and further experiments about our work in appendices:

- Sec.A: the summarized **algorithm** of the proposed method.

- Sec.B: the full **proof** of Proposition 1.

- Sec.C: the full **proof** of Theorem 1.

- Sec.D: the full **proof** of Theorem 2.

- Sec.E: the detailed discussion about the **space-time complexity** comparing with other FL methods.

- Sec.E: the detailed discussion about more **related works**.

- Sec.G: the **implementation details** of experiments such as datasets, baselines, models, hyper-parameters and sparse model aggregation procedure of pFedGate.

- Sec.H: further **experimental results** including: Sec.H.1, the performance in novel client participation case; Sec.H.2, the performance in partial clients participation case; Sec.H.3, the convergence study; and Sec.H.4, the model compression manner study, in which we compare pFedGate to the SOTA binary quantization method FedMask (Li et al., 2021a).

## A    THE ALGORITHM OF THE PROPOSED PFEDGATE METHOD

---

**Algorithm 1:** Efficient Personalized FL with pFedGate

**Input:** $T, \eta_g, \eta, \theta_g^0, \{\phi_i^0\}_{i \in \mathcal{C}}, s_{min}, \{s_i\}_{i \in \mathcal{C}}$

1 **for** $t = 1, \cdots, T$ **do**
2      Server sends $\theta_g^{t-1}$ to clients $\mathcal{C}_s$ sampled from $\mathcal{C}$
3      **for** *client $i \in \mathcal{C}_s$ in parallel* **do**
4          **for** *data sample $(x, y) \in \mathcal{S}_i$* **do**
5              Get sparse gated weight: $M_i' = g(\phi_i^{t-1}; x)$
6              Do personalized model-adaption: $\theta_i' = \theta_g^{t-1} \circledast M_i'$
7              Update global model and local gating layer: $\theta_g^t = \theta_g^{t-1} - \eta_g(\nabla_\theta f(\theta_i'; x, y))$
8              $\phi_i^t = \phi_i^{t-1} - \eta(\nabla_\phi \theta_i' \nabla_\theta f(\theta_i'; x, y))$
9          Upload sparse $\Delta\theta_{g,i}^t := \theta_g^t - \theta_g^{t-1}$
10      Server updates global model: $\theta_g^t = \text{AGGREGATE}\left(\theta_g^{t-1}, \{\Delta\theta_{g,i}^t\}_{i \in \mathcal{C}_s}\right)$
11 **return** $\theta_g, \{\phi_i\}_{i \in \mathcal{C}}$

---

We summarize the overall algorithm in Algorithm 1. Besides, we present more details about the gradients flow through the gating layer, which contains a knapsack solver.

We leverage the straight-through trick (Hubara et al., 2016) to make the combination of the two flows predicted by the gating layer (red $G$ and blue $M$ in Figure 2) still differentiable. Recall that the combination is $M' = M \cdot I^*$ where $I^*$ is a binary index and a solution to the knapsack problem (Section 5.2). The "straight-through" (ST) here means that we build such a computation graph, in which we only use binary variable $I^*$ in the forward pass stage, while *replace $I^*$ into the differentiable variable $G$ in backward gradient propagation*. Specifically, this can be implemented by the following demonstrative PyTorch code: $I_{ST}^* = I^*$ - $G$.detach() + $G$. In this way, the tensor $I_{ST}^*$ and $M$ are both differentiable and can be used to pass the gradient flow via their combination tensor $M' = I_{ST}^* * M$ and the following adapted tensor of the sparse model $\theta_i' = \theta_g^{t-1} \circledast M_i'$. And thus we can train the personalized model and the gating layer as Algorithm 1, lines 6 and 7 shown.

## B  PROOF OF PROPOSITION 1

Since $f$ is $\mu$-strongly convex, by definition we have

$$f(\theta') \geq f(\theta) + \nabla_\theta f(\theta)^T (\theta' - \theta) + \frac{\mu}{2} \|\theta' - \theta\|^2, \ \forall \theta, \theta'. \tag{8}$$

We have $f'(\theta) = f(\theta) - \frac{\mu}{2} \|\theta\|^2$ is also convex due to the first order condition of $f$ with Equation (8). Considering the monotone gradient condition for convexity of $f'$, we get

$$(\nabla_\theta f(\theta) - \nabla_\theta f(\theta'))^T (\theta - \theta') \geq \mu \|\theta - \theta'\|^2, \ \forall \theta, \theta'. \tag{9}$$

Applying Cauchy-Schwartz inequality on Equation (9), we get

$$\begin{aligned}
\|\nabla_\theta f(\theta) - \nabla_\theta f(\theta')\| \|\theta - \theta'\| \\
\geq (\nabla_\theta f(\theta) - \nabla_\theta f(\theta')^T (\theta - \theta') \\
\geq \mu \|\theta - \theta'\|^2.
\end{aligned} \tag{10}$$

Dividing $\|\theta - \theta'\|$ on both sides of Equation (10), we get

$$\|\theta - \theta'\| \leq \|\nabla_\theta f(\theta) - \nabla_\theta f(\theta')\| / \mu, \ \forall \theta, \theta'. \tag{11}$$

Under Assumption 1, for all $(\theta_i^*, \theta_g^*)$ pairs with Equation (11), we have

$$\|(M_i^* - 1)\theta_g^*\|^2 \leq \cdot \frac{\sigma_i^2}{\mu^2}, \quad \forall i \in \mathcal{C}. \tag{12}$$

Using the reversed Cauchy-Schwarz inequality (Hardy et al., 1952), we have

$$A_i \cdot \|(M_i^* - 1)\theta_g^*\|^2 \geq \|(M_i^* - 1)\| \|\theta_g^*\|, \tag{13}$$

where $A_i = \frac{1}{4} \frac{(R_{\theta_i} R_{\theta_g} + r_{\theta_i} r_{\theta_g})^2}{R_{\theta_i} R_{\theta_g} r_{\theta_i} r_{\theta_g}}$ is a positive constant, $r_{\theta_i}$ and $r_{\theta_g}$ are the infimum of $\theta_i$ and $\theta_g$ respectively, $R_{\theta_i}$ and $R_{\theta_g}$ are the supremum of $\theta_i$ and $\theta_g$ respectively. Since $R_{\theta_i} R_{\theta_g} \geq r_{\theta_i} r_{\theta_g}$, we have $A_i \leq \frac{1}{4} \frac{(2 R_{\theta_i} R_{\theta_g})^2}{R_{\theta_i} R_{\theta_g} r_{\theta_i} r_{\theta_g}} = \frac{R_{\theta_i} R_{\theta_g}}{r_{\theta_i} r_{\theta_g}}$. Combining Equation (12) and Equation (13), we get the upper bound in Propostion 1 and finish the proof.

## C  PROOF OF THEOREM 1

Let $\mathbb{H}^n$ denote the function space with its elements parametrized by $\theta_g, \phi_1, \cdots, \phi_n$ and the distance metric is defined as:

$$\begin{aligned}
d((\theta_g, \phi_1, \cdots, \phi_n) - (\theta_g', \phi_1', \cdots, \phi_n')) \\
= \frac{1}{n} \mathbb{E}_{x,y \sim \mathcal{D}_i} \left[ \left| \sum (f(\theta_g, \phi_i; x, y) - \sum f(\theta_g', \phi_i'; x, y) \right| \right],
\end{aligned} \tag{14}$$

With the Lipshitz conditions in Assumption 2, we have:

$$\begin{aligned}
& d((\theta_g, \phi_1, \cdots, \phi_n) - (\theta_g', \phi_1', \cdots, \phi_n')) \\
& \leq \sum_i \frac{1}{n} \mathbb{E}_{x,y \sim \mathcal{D}_i} \left[ \ell(h_{\theta_g, \phi_i}(x), y) - \ell(h_{\theta_g', \phi_i'}(x), y) \right] \\
& \leq L_f \|h_{\theta_g, \phi_i} - h_{\theta_g', \phi_i'}\| \\
& = L_f \|h_s(\theta_i) - h_s(\theta_i')\| \\
& \leq L_f L_h \|\theta_g M_i - \theta_g' M_i'\| && \text{(Lipshitz condition)} \\
& \leq L_f L_h \|\theta_g M_i - \theta_g M_i' + \theta_g M_i' - \theta_g' M_i'\| \\
& \leq L_f L_h \left[ \|\theta_g\| \cdot \|M_i - M_i'\| + \|M_i'\| \cdot \|\theta_g - \theta_g'\| \right] \\
& \leq L_f L_h \left[ R \|M_i - M_i'\| + \|\theta_g - \theta_g'\| \right] && \text{(Assumption 2)} \\
& \leq L_f L_h \left[ R \|g(\phi_i) - g(\phi_i')\| + \|\theta_g - \theta_g'\| \right] \\
& \leq L_f L_h \left[ R L_g \|\phi_i - \phi_i'\| + \|\theta_g - \theta_g'\| \right], && \text{(Lipshitz condition)}
\end{aligned} \tag{15}$$

 We can get an $\epsilon$-covering in metric $d((\theta_g, \phi_1, \cdots, \phi_n) - (\theta_g', \phi_1', \cdots, \phi_n'))$ if we select a covering of in the parameter space with both $||\phi_i - \phi_i'||$ and $||\theta_g - \theta_g'||$ equal to $\frac{\epsilon}{L_f L_g (R L_g + 1)}$. Therefore, the covering number of $\mathbb{H}^{|\mathcal{C}|}$, denoted as $\mathcal{B}(\epsilon, \mathbb{H}^{|\mathcal{C}|})$ is: $\log(\mathcal{B}(\epsilon, \mathbb{H}^{|\mathcal{C}|})) = \mathcal{O}\left((|\mathcal{C}|d_\phi + d) \log \frac{R L_f L_h (R L_g + 1)}{\epsilon}\right)$.

According to Baxter (2000); Shamsian et al. (2021), there exit $\tilde{N}$ and $\tilde{N} = \mathcal{O}\left(\frac{1}{n\epsilon^2} \log \frac{\mathcal{B}(\epsilon, \mathbb{H}^{|\mathcal{C}|})}{\delta}\right) = \mathcal{O}\left(\frac{d}{|\mathcal{C}|\epsilon^2} \log \frac{R L_f L_h (R L_g + 1)}{\epsilon} + \frac{d_\phi}{\epsilon^2} \log \frac{R L_f L_h (R L_g + 1)}{\epsilon} - \frac{\log \delta}{|\mathcal{C}|\epsilon^2}\right)$.

## D    PROOF OF THEOREM 2

We note that Marfoq et al. (2021) extends the surrogate optimization into FL setting and provide convergence results for the algorithms which approximates the target objective functions using first-order surrogate or partial first-order surrogate functions. For simplicity, we use two more compact notations for all the learnable parameters introduced in our formulation as $\vec{v}_{M'} \triangleq \{M_i\}_{i \in \mathcal{C}}$ and $\vec{v}_{M', \phi} \triangleq \{M_i' = g(\phi_i; x)\}_{i \in \mathcal{C}, x \in X_i}$. Here we prove that our method described in Algorithm 1 can be regarded as to optimize the objective function $F(\theta_g, \vec{v}_{M'})$ in Equation (5) with *another partial first-order surrogate function*, and the convergence results from Marfoq et al. (2021) can be applied into our method. We first recap the formal definition of partial first-order surrogate:

**Definition 1** (Partial first-order surrogate Marfoq et al. (2021)). *A function $F'(\vec{u}, \vec{v}) : \mathbb{R}^{d_u} \times \mathcal{V} \to \mathbb{R}$ is a partial-first-order surrogate of $F(\vec{u}, \vec{v})$ wrt $\vec{u}$ near $(\vec{u}_0, \vec{v}_0) \in R^{d_u} \times \mathcal{V}$ when the following conditions are satisfied:*

1. *$F'(\vec{u}, \vec{v}) \geq F(\vec{u}, \vec{v})$ for all $\vec{u} \in \mathbb{R}^{d_u}$ and $\vec{v} \in \mathcal{V}$;*

2. *$e(\vec{u}, \vec{v}) \triangleq F'(\vec{u}, \vec{v}) - F(\vec{u}, \vec{v})$ is differentiable and L-smooth with respect to $\vec{u}$. Moreover, we have $e(\vec{u}_0, \vec{v}_0) = 0$ and $\nabla_{\vec{u}} e(\vec{u}_0, \vec{v}_0) = 0$.*

3. *$F'(\vec{u}, \vec{v}_0) - F'(\vec{u}, \vec{v}) = d_{\mathcal{V}}(\vec{v}_0, \vec{v})$ for all $\vec{u} \in \mathbb{R}^{d_u}$ and $\vec{v} \in \arg\min_{\vec{v}' \in \mathcal{V}} F'(\vec{u}, \vec{v}')$, where $d_{\mathcal{V}}$ is non-negative and $d_{\mathcal{V}}(\vec{v}, \vec{v}') = 0 \iff \vec{v} = \vec{v}'$.*

Definition 1 indicates that in partial parameter set $\mathcal{V}$, $F'$ is *majorant* to $F$ and the *smoothness* is satisfied in the neighborhood of given point.

Denote $F_i'(\theta_g, \vec{v}_{M', \phi})$ as the transformed local objective function depicted by the gating layer $\phi_i$, *i.e.*, the sparse gated weight $M_i'$ is generated by the process described in Section 5.2 and intermediate objective in Equation (6). We then restrict our attention from the function $F(\theta_g, \vec{v}_{M'})$ over all clients into $|\mathcal{C}|$ local functions $F_i(\theta_g, \vec{v}_{M', \phi})$ for each client $i$, since the weighted sum operation is convex and holds the partial majorization and smoothness properties. Consider the iterative version of $F_i$ and $F_i'$ corresponding to the produce in Algorithim 1. We have the partial set of gated weights $\mathcal{V}_\phi = \{\{M_i' = g(\phi_i^t; x)\}_{x \in X_i}\}_{\phi_i^t = \arg\min_\phi (F_i'(\theta_g^{t-1}, \vec{v}_\phi^{t-1}))}$ for all $t \in [T]$. For $\vec{v}_{M', \phi_i^t} \in \mathcal{V}_\phi$, due to the optimality of $\phi_i^t$ and equality between $\vec{v}_{M'}$ and $\vec{v}_{M', \phi_i^t}$ using block-to-element scatter operation on the gated weights, we have $F_i'(\theta_g, \vec{v}_{M'}) = F_i(\theta_g, \vec{v}_{M', \phi_i^t})$ and Condition 1 satisfied.

Note that the difference between $F_i'(\theta_g, \vec{v}_{M'})$ and $F_i(\theta_g, \vec{v}_{M', \phi})$ is differentiable w.r.t. $\theta_g$ since both of them is differentiable by adopting either element-wise or block-wise production on $\theta_g$ and the sparse gated weights. With the smoothness Assumption 3, we have the Condition 2 satisfied. For the Condition 3, it is clear that if the right statement holds, *i.e.*, $\vec{v}_{M', \phi} = \vec{v}_{M', \phi}'$, then $d_{\mathcal{V}}(\vec{v}_{M', \phi}, \vec{v}_{M', \phi}') = F_i'(\theta_g, \vec{v}_{M', \phi}) - F_i'(\theta_g, \vec{v}_{M', \phi}') = 0$ and we have the sufficiency between the two statements of Condition 3 satisfied. For the necessity between the two statements of Condition 3, we can consider such a set $\mathcal{V}_\phi \setminus (\tilde{\mathcal{V}}_\phi \cup \overline{\mathcal{V}}_\phi)$, where $\tilde{\mathcal{V}}_\phi$ indicates the subset of $\mathcal{V}_\phi$ in which $t \in [\tilde{T}, T]$ if $F_i'$ converges at step $\tilde{T} \leq T$, and $\overline{\mathcal{V}}_\phi$ indicates the subset of $\mathcal{V}_\phi$ in which $t \in \{t' | F_i'(\theta_g^{t-1}, \vec{v}_\phi^{t-1}) = F_i'(\theta_g^{t'-1}, \vec{v}_\phi^{t'-1}), \forall t, t' \in [T], t' \neq t\}$. The new partial subset ensures the optimal $\vec{v}_{M', \phi}$ is unique by discarding the redundant $\vec{v}_{M', \phi}$ that have the same $F_i'(\theta_g, \vec{v}_{M', \phi})$. For

such a new partial subset $\mathcal{V}_\phi \setminus (\tilde{\mathcal{V}}_\phi \cup \overline{\mathcal{V}}_\phi)$, we have the necessity between the two statements of Condition 3 satisfied due to the uniqueness of the solutions of $\arg\min_{\vec{v}'_{M',\phi} \in \mathcal{V}_\phi \setminus (\tilde{\mathcal{V}}_\phi \cup \overline{\mathcal{V}}_\phi)} F'(\vec{\theta_g}, \vec{v}'_{M',\phi})$,

and the Condition 1 and Condition 2 still hold since $\mathcal{V}_\phi \setminus (\tilde{\mathcal{V}}_\phi \cup \overline{\mathcal{V}}_\phi)$ is a subset of $\mathcal{V}_\phi$.

We finally show that the assumptions required by the convergence proof in Marfoq et al. (2021) hold when using our approach.

**Proposition 2.** *Under the assumptions 1–5, when taking the parameter from $\theta_g$ to $\theta_{\phi_i} = \theta_g \circledast \phi_i(x)$ over all $i \in \mathcal{C}, x \in X_i$, the gradient variance of $f$ and diversity across clients are also bounded, and the smoothness of $f$ holds.*

*Proof.* Note that using the chain rule, the gradients on $\phi_i$ can be written as $\nabla_\phi(f(\theta_{\phi_i}; x, y)) = \nabla_\phi \theta_{\phi_i} \nabla_\theta f(\theta_{\phi_i}; x, y)$ for $(x, y) \in \mathcal{D}_i$. For simplicity, we denote $\nabla_\phi(f(\theta_{\phi_i}; x, y)) \triangleq f(\theta_{\phi_i})$. Since we generate sparse model as $\theta_{\phi_i} = \theta_g \circledast M'_i$, we have $\nabla_\phi \theta_{\phi_i} = M'_i \nabla_\phi M'_i$. The block-wise production operation $\circledast$ is differentiable and $f$ over $\theta_g$ is differentiable according to Assumption 3, thus $f$ over $\phi_i$ is also differentiable. Then recall that the optimal gated weights are bounded as shown in Proposition 1. Further, we explicitly bound the sparse gated weight during FL training process in $[0, 1]^L$ using sigmoid function as introduced in Section 5.2. We thus can assume that

$$||\nabla_\phi \theta_{\phi'_i} - \nabla_\phi \theta_{\phi_i}|| \le L_\phi ||\theta_{\phi'_i} - \theta_{\phi_i}||^2, \tag{16}$$

where $L_\phi$ is a Lipshitz constant corresponding to our bounded gated weights. We get

$$
\begin{aligned}
&||\nabla_\phi f(\theta_{\phi'_i}) - \nabla_\phi f(\theta_{\phi_i})||^2 \\
=&||\nabla_\phi \theta_{\phi'_i} \nabla_\theta f(\theta_{\phi'_i}) - \nabla_\phi \theta_{\phi_i} \nabla_\theta f(\theta_{\phi_i})||^2 \qquad\qquad \text{\color{red}(Chain Rule)} \\
=&||\nabla_\phi \theta_{\phi'_i} \nabla_\theta f(\theta_{\phi'_i}) - \nabla_\phi \theta_{\phi'_i} \nabla_\theta f(\theta_{\phi_i}) + \nabla_\phi \theta_{\phi'_i} \nabla_\theta f(\theta_{\phi_i}) - \nabla_\phi \theta_{\phi_i} \nabla_\theta f(\theta_{\phi_i})||^2 \\
\le&||\nabla_\phi \theta_{\phi'_i} \nabla_\theta f(\theta_{\phi'_i}) - \nabla_\phi \theta_{\phi'_i} \nabla_\theta f(\theta_{\phi_i})||^2 + ||\nabla_\phi \theta_{\phi'_i} \nabla_\theta f(\theta_{\phi_i}) - \nabla_\phi \theta_{\phi_i} \nabla_\theta f(\theta_{\phi_i})||^2 \\
\le& L||\nabla_\phi \theta_{\phi'_i}||^2 ||\theta_{\phi'_i} - \theta_{\phi_i}||^2 + L_\phi ||\nabla_\theta f(\theta_{\phi_i})||^2 ||\theta_{\phi'_i} - \theta_{\phi_i}||^2 \qquad \text{\color{red}(Assumption 3 \& Eqn. (16))} \\
\le& 2LL_\phi ||\theta_{\phi'_i} - \theta_{\phi_i}||^2,
\end{aligned}
\tag{17}
$$

where the last two lines is because of that Lipschitz implies bounded gradient. Thus we get that $f$ is $(2LL_\phi)$-smooth using our method. Similar to the proof of smoothness that leverages the chain rule and bounded $M_i$, we can generalize Assumption 1 and Assumption 5 for our method such that the gradients dissimilarity is bounded and $f$ has bounded variance characterized by another constant factor. $\qquad\square$

Now we see that each client can optimize a partial first-order surrogate function and the necessary assumptions satisfy with Proposition 2. Applying the Theorem 3.2' in Marfoq et al. (2021), we conclude the proof of our Theorem 2.

**Remark on the assumptions.** We note that the adopted theoretical assumptions are widely used by numerous related FL works. Specifically, the assumptions 1 (bounded data diversity), 3 (smoothness of $f$), 4 (bounded output of $f$), 5 (bounded gradients variance) are used in works such as Li et al. (2019); Ma et al. (2022); Dinh et al. (2020b); Liang et al. (2020); Marfoq et al. (2021). These assumptions are reasonable and fairly mild in FL scenarios:

- The assumptions 3 and 4 are easily satisfied for most discriminative models such as neural networks (Tan et al., 2021; Kairouz et al., 2021; Li et al., 2019).

- The assumptions 1 and 5 are necessary to a feasible and effective FL course, in which there is knowledge that can be shared and mutually beneficial between the FL participants (Ma et al., 2022; Dinh et al., 2020b; Liang et al., 2020; Marfoq et al., 2021).

- As for the assumption 2 (bounded parameters of the global model and the gating layer), it is dependent on the hypothesis space of the adopted global model and gating layer, which are controllable by users. For example, we can hard clip their model parameters, and introduce bounded activation functions such as sigmoid into the gating layer structure as the proposed method does. A

more specific example of this is our implementation, where we use zero initialization, kaiming-uniform initialization and (0, 1)-uniform initialization for the parameters of adopted global model and gating layer, which leads to all these init parameters are bounded by one of 0, 1, or constants calculated by the bounded *fan_in* (usually determined by the data shape). Then during the FL courses, we clip the gradients at each backpropagation step (via *torch.nn.utils.clip_grad_norm_()*), resulting in the bounded parameters and thus the assumption holds.

- Besides, assumption similar to our assumption 2 can be found in the related pFL work, pFedHN (Shamsian et al., 2021), in which the authors assume that their introduced weights of the hyper-network and the embeddings are bounded and follow several Lipschitz conditions with constants different from ours.

## E  SPACE-TIME COMPLEXITY

Table 3: The proposed pFedGate achieves better computation, communication and storage complexity over state-of-the-art personalized FL methods. The underlined results indicate the examples when taking default hyper-pameters: the embedding and hypernetwork size $d_v = 0.25|\mathcal{C}|, d_h = 100$ for pFedHN Shamsian et al. (2021), the component model number $k = 3$ for FedEM Marfoq et al. (2021), the client sparsity $s_i = 0.5$, and relative sparsity of gating layer $s_{\phi_i} = d/2d_X L = 0.05$ for pFedGate. Besides, for pFedGate, we report the average non-zero parameter ratio $q_i = 0.67$, which is dependent on datasets and local training steps. Here we report the average value of $q_i$ when we run FL experiments 3 times on FEMNIST and CIFAR10 with $s_i = 0.5$, local update step as 1 epoch, and batch size as 128.

| | | FedAvg | pFedHN | FedEM | pFedGate |
|---|---|---|---|---|---|
| Computation | Train (Client) | $\mathcal{O}(2d)$ | $\mathcal{O}(2d)$ | $\mathcal{O}(2kd)$ $\mathcal{O}(6d)$ | $\mathcal{O}\big(2d(s_i + s_{\phi_i})\big)$ $\mathcal{O}(1.1d)$ |
| | Infer (Client) | $\mathcal{O}(d)$ | $\mathcal{O}(d)$ | $\mathcal{O}(d)$ | $\mathcal{O}\big(d(s_i + s_{\phi_i})\big)$ $\mathcal{O}(0.55d)$ |
| | Train (Server) | $\mathcal{O}(1)$ $\mathcal{O}(6d)$ | $\mathcal{O}\big((d_v + d_h)|\mathcal{C}_s|\big)$ $\mathcal{O}\big((0.25|\mathcal{C}|+100)|\mathcal{C}_s|\big)$ | $\mathcal{O}(1)$ | $\mathcal{O}(1)$ |
| Communication | Client | $\mathcal{O}(d)$ | $\mathcal{O}(d)$ | $\mathcal{O}(kd)$ $\mathcal{O}(3d)$ | $\mathcal{O}(q_i d)$ $\mathcal{O}(0.67d)$ |
| | Server | $\mathcal{O}(d|\mathcal{C}_s|)$ | $\mathcal{O}(d|\mathcal{C}_s|)$ | $\mathcal{O}(kd|\mathcal{C}_s|)$ $\mathcal{O}(3d|\mathcal{C}_s|)$ | $\mathcal{O}(d|\mathcal{C}_s|)$ |
| Storage | Client | $\mathcal{O}(d)$ | $\mathcal{O}(d)$ | $\mathcal{O}(kd)$ $\mathcal{O}(3d)$ | $\mathcal{O}\big((1 + s_{\phi_i})d\big)$ $\mathcal{O}(1.05d)$ |
| | Server | $\mathcal{O}(d)$ | $\mathcal{O}(d_v d_h + d_v|\mathcal{C}|)$ $\mathcal{O}(25|\mathcal{C}|+0.25|\mathcal{C}|^2)$ | $\mathcal{O}(kd)$ $\mathcal{O}(3d)$ | $\mathcal{O}(d)$ |

We now examine the efficiency of pFedGate described in Algorithm 1 in terms of computation, communication, and storage costs. We will see that pFedGate gains improvements on computation and communication costs with a tolerable storage cost. Table 3 summarizes the costs for pFedGate and competitors including FedAvg and two SOTA personalized FL methods, where the training and inference costs on client is considered for each training batch (or each sample when the batch size is 1), and the communication cost is considered for each FL round. We also mark some specific example values when taking the default hyper-parameters of different methods in Table 3. For comparison simplicity, here we assume that for the computation costs, the inference and back-propagation times are proportional to the model size. We hide both constants and polylogarithmic factors dependent on the specific model architectures in $\mathcal{O}(\cdot)$, since we compare these methods with the same model architectures and the same trained data samples. As for communication and storage, $\mathcal{O}(\cdot)$ hides constants and lower-order factors dependent on specific implementations such as platforms, storage, and communication protocols. The pFedGate uploads parameters in $\mathcal{O}(q_i d)$, where $q_i = count(\Delta\theta_i \neq 0)/d$ indicates the ratio of non-zero model updates. Specifically, $q_i$ can be

regarded as the probability that a parameter is finally not masked in the local training process. We can consider $q_i = \left(1 - \prod_{j=1}^{|\mathcal{S}|}(1 - p_j)\right)$, where $p_j$ indicates the probability that a parameter is not within the sub-blocks the gating layer predicts to mask when taking the $j$-th data batch as input, and $|\mathcal{S}|$ indicates the number of data batches trained in each local FL round. Note that $p_1$ is equal to the sparsity factor $s_i$ at client $i$, and $p_j$ becomes smaller as $j$ increases, due to the fact that the local data samples usually share some similarities, and the newly predicted sub-blocks are more likely to be the same as the sub-blocks have been selected.

From Table 3 we can see that pFedGate achieves better efficiency than FedAvg in training, inference, and uploading with negligible additional storage cost on clients (as marked in red). By contrast, to achieve good personalization, FedEM introduces more than one global model and pays larger costs in training, communication, and storage (as marked in blue). And pFedHN learns client-wise embedding in the server, which may lead to a single-point bottleneck for cross-device setting and potentially disclose personal information.

## F  MORE RELATED WORK COMPARISON

Here we present more details about the works related to our method. QuPeD (Ozkara et al., 2021) combines quantization and knowledge distillation for personalized federated learning. FedMask (Li et al., 2021a) freezes the local models during the whole FL process, learns distinct binary masks for the last several layers of the models, and only transmits and aggregates the learned masks. Different from the quantization based methods, we adopts continuous gated weights on all sub-blocks of the local model and transmits the sparse model updates, enabling flexible model size reduction and high capability to capture the client relationships via the shared global model.

FedRep (Collins et al., 2021) proposes to upload partial sub-parameters as shared representation and leave the others locally trained. HereroFL (Diao et al., 2021) selects different subsets of global model parameters for clients based on their computational capabilities. Different from these model-decoupling based works that use the same parameter subset for those clients having the same computation capability, our method generates different sub-models from the whole global model with a larger learnable parameter space to handle heterogeneous data distribution. More importantly, our method differs from these works by adaptively generating the sparse model weights at a fine-grained sample level, which achieves a good estimation for the conditional probability of heterogeneous local data and high accuracy as shown in our experiments.

FedPNAS Minh & Carl (2021) leverages neural architecture search (NAS) to search suitable sub-networks for clients, and FedMN Tianchun et al. (2022) proposes to use routing-hypernetwork to select personalized sub-blocks. For the FedPNAS, we differ from it in the target and studied problem. FedPNAS mainly focuses on improving the personalization performance with sufficient hardware resources. For example, they adopt 5 clients in their experiments that usually correspond to the cross-silo case. While pFedGate focuses on the cross-device setting, where the clients' resources are usually very limited. Compared with FedMN, the performance of pFedGate is theoretically guaranteed for both generalization and convergence. While FedMN doesn't provide any theoretical analyses for generalization or convergence. Besides, the proposed pFedGate method adopts a much larger hypotheses space than FedMN. In FedMN, each modular is either connected or dropped. While pFedGate doesn't only decide the connections of network (connected or dropped), but also scales weights of the remained blocks simultaneously to improve the personalized performance.

## G  IMPLEMENTATION DETAILS

**Datasets.**  We conduct experiments on several widely used FL datasets including EMNIST Cohen et al. (2017) and FEMNIST Caldas et al. (2018) for 62-class hadnwritten character recognition, and CIFAR10/CIFAR100 Krizhevsky (2009) for 10-class/100-class image classification. We follow the heterogeneous partition manners used in Marfoq et al. (2021); Caldas et al. (2018); Diao et al. (2021); Fallah et al. (2020): FEMNIST was partitioned by writer and we generate the other three FL datasets using Dirichlet allocation of parameter $\alpha$=0.4. We adopt 10% sub-samples of EMNIST (81,425 samples) and 15% sub-samples of FEMNIST (98,671 samples), and allocate the sub datasets to 100

and 539 clients for EMNIST and FEMNIST respectively. For CIFAR10/CIFAR100 (60,000 samples), we allocate them to 100/50 clients. All datasets are randomly split into train/valid/test sets with ratio 6:2:2.

**Baselines and models.** We consider the following competitive baselines including SOTA personalized and efficient FL methods in this work:

- *FedAvg* McMahan et al. (2017): the classical FL method that simply aggregates model updates in a weighted averaging manner;

- *FedAvg-FT*: a basic personalised baseline that fine-tunes the global model with local data before model local evaluation;

- *Local*: a naive personalized method that trains models on local datasets without FL communication;

- *pFedMe* Dinh et al. (2020b) decouples the personalized model and global model with Moreau envelops based regularization;

- *LG-FedAvg* Liang et al. (2020) achieves personalization with improved communication efficiency;

- *Ditto* Li et al. (2021b) is a SOTA pFL method that introduces a local personalized model for each client, which is trained with model parameter regularization according to global model;

- *FedEM* Marfoq et al. (2021) deals with data heterogeneity via the mixture of multiple global models. We use 3 models according to the authors' default choice;

- *FedRep* Marfoq et al. (2021) proposes to improve the FL performance and communication efficiency via sharing partial model parameters among FL participants. Only the low-layers of models for feature extraction are uploaded to server and aggregated, and the remaining head parameters are locally trained for personalization. Following the authors' sharing manner, we adopt the last classification layers as local personalized model parameter;

- *FedMask* Li et al. (2021a) learns distinct binary masks for the last several layers of the local models, and aggregates the masks with an intersection operation. Following the mask layer choice manner similar to the authors, in our experiments, the one-shot mask is applied in the last layer for the adopted 2-layer CNN and the last 2 layers for the adopted 3-layer CNN respectively;

- *HeteroFL* Diao et al. (2021) uses heterogeneous model architectures for different clients to achieve personalization and improvements in both computational and communication efficiency. Recall that we can vary each client's sparse ratio $s$ to reflect the clients' resource heterogeneity, since the affordable sparse ratio $s$ can fairly reflect the client's computation and communication capabilities. Clients with very limited computation and communication resources have to set $s$ close to 0; On the contrary, clients with sufficient resources can set $s$ close to 1. Here we use the full model parameters for 10% clients and 50% parameters for the other 90% clients following the authors' computation complexity setting, which leads to an average sparsity $\bar{s}' = 0.55$.

To align with previous works, we use a 2-layer CNN Reddi et al. (2021); Marfoq et al. (2021) for EMNIST/FEMNIST, and two LeNet-based CNNs Dinh et al. (2020b); Liang et al. (2020); Shamsian et al. (2021) with different capabilities for CIFAR10 and CIFAR100. Specifically, the model used for EMNIST/FEMNIST two convolutional layers with $5 \times 5$ kernels, max pooling and two dense layers with a total of 2,171,786 parameters. The models used for CIFAR10 and CIFAR100 have a LeNet-based structure with an additional linear classification layer, and a total of 256,830 and 3,537,444 parameters respectively.

**Platform and Hyper-parameters.** We implement all models with PyTorch, and run experiments on Tesla V100 and NVIDIA GeForce GTX 1080 Ti GPUs. For fair comparisons, we adopt the same communication rounds for all methods and search for the optimal configuration of hyper-parameters using

Table 4: The adopted learning rates for pFedGate in all datasets.

| | EMNIST | | FEMNIST | | CIFAR10 | | CIFAR100 | |
|---|---|---|---|---|---|---|---|---|
| | $\eta_g$ | $\eta$ | $\eta_g$ | $\eta$ | $\eta_g$ | $\eta$ | $\eta_g$ | $\eta$ |
| $s$=1 | 0.1 | 0.1 | 0.1 | 0.1 | 0.03 | 0.05 | 0.1 | 1.5 |
| $s$=0.5 | 0.1 | 0.1 | 0.1 | 0.05 | 0.03 | 1.5 | 0.03 | 1.5 |
| $s$=0.3 | 0.03 | 0.5 | 0.3 | 0.05 | 0.03 | 1.5 | 0.05 | 0.5 |

the validation sets. We run each experiment 3 times with the optimal configurations and different random seeds, and report the average results. For each method on each dataset, we use the SGD optimizer and grid search the learning rate $\eta_g$ from $[0.005, 0.01, 0.03, 0.05, 0.1, 0.3, 0.5]$, set the communication round $T = 400$, the batch size as 128 and the local update step as 1 epoch. For pFedGate, the learning rate of gating layer $\eta$ is searched from $[0.01, 0.05, 0.1, 0.3, 0.5, 1, 1.5]$, and we set the block size splitting factor $B = 5$ for all evaluated models. For FedAvg-FT, we use the local fine-tuning step as 1 epoch. For pFedMe, we search its penalization parameter $\mu$ from $[0.0001, 0.001, 0.01, 0.1, 1, 10]$. For Ditto, we search its regularization factor from $[0.05, 0.1, 0.5, 0.8]$. For FedRep, we search its learning rate for personalized model from $[0.05, 0.005, 0.5, 0.01, 0.1]$. For FedEM, we set its number of global models as 3. We summarize the adopted learning rates in Table 4.

**Global Model Aggregation Detail.** We present more details about the global model aggregation of pFedGate. After local training, the client only uploads the updates of the parameters in the global model that are selected to form the client's local model. In other words, the client uploads the *non-zero* parameters updates with their corresponding *index* in the global model. The server then aggregates the received model updates according to their index.

Considering the following toy example: there are four clients, and the global model has a total of three parameter blocks. Each client uploads a dictionary whose key denotes the index and the value denotes the corresponding parameter updates.

- Client 1: 0: $\delta_0$, 2: $\delta_2$;

- Client 2: 1: $\delta_0$, 2: $\delta_2$;

- Client 3: 0: $\delta_0$, 1: $\delta_2$;

- Client 4: 0: $\delta_0$, 1: $\delta_0$, 2: $\delta_2$.

After receiving the above updates, the server aggregates the parameter updates with index 0 by weighted averaging the $\delta_0$s from clients 1, 3, and 4. The other parameters are aggregated similarly.

# H  ADDITIONAL EXPERIMENTS

## H.1  NOVEL CLIENTS GENERALIZATION

To evaluate the generalization of pFedGate for the novel clients that haven't participated in the previous FL training stage, we conduct experiments with similar simulation settings to Diao et al. (2021); Marfoq et al. (2021); Yuan et al. (2022), where the novel clients only fine tune the global model on their local dataset. Specifically, we randomly select 20% clients as novel clients, and evaluate the selected and remaining clients separately. The results are summarized in Table 5, where $\overline{Acc}$ and $\widetilde{Acc}$ indicate the results of FL-participated clients and novel clients respectively, and $\Delta = \overline{Acc} - \widetilde{Acc}$ indicates the accuracy generalization gap. The results show that the proposed method pFedGate can generalize to novel clients well and gain smaller accuracy gaps than compared methods. For the novel clients, instead of re-training the over-parameterized global model on (possibly small amounts of) local data, pFedGate only needs to train the personalized lightweight gating layers with a small number of parameters, and thus achieves an efficient and effective local adaptation.

Table 5: Averaged accuracy when testing on novel clients that are not participated in the training stage of FL. The $\widetilde{Acc}$ indicates the results of novel clients, and $\Delta$ indicates the generalization gap $\overline{Acc} - \widetilde{Acc}$.

| | FEMNIST | | | CIFAR10 | | |
| | $\overline{Acc}$ | $\widetilde{Acc}$ | $\Delta$ | $\overline{Acc}$ | $\widetilde{Acc}$ | $\Delta$ |
|---|---|---|---|---|---|---|
| FedAvg | 76.39 | 74.93 | 1.46 | 67.78 | 66.15 | 1.63 |
| FedEM | 79.21 | 78.26 | 0.95 | 71.92 | 70.75 | 1.17 |
| pFedGate, s=1 | 87.79 | 87.18 | 0.61 | 73.97 | 73.18 | 0.79 |
| pFedGate, s=0.5 | 86.64 | 86.19 | 0.45 | 74.12 | 73.19 | 0.93 |
| pFedGate, s=0.3 | 85.69 | 85.12 | 0.57 | 73.64 | 72.78 | 0.86 |

## H.2 PARTIAL CLIENTS PARTICIPATION

We have seen our method can consistently and simultaneously improve the efficiency and model accuracy as Table 1 shows. One can arise such a natural question: when combined with other efficiency improvement techniques such as client sampling, is pFedGate still effective? Here we run experiments by uniformly sampling 20% clients without replacement at each round, and summarize the results in Table 6. It is observed that compared with baselines having about 1.45% $\sim$ 3.56% performance drop, our method still achieves strong and comparable performance to the non-sampled case, verifying the effectiveness and robustness of our proposed method again.

## H.3 CONVERGENCE STUDY

We demonstrate the convergence curves of FedAvg, HeteroFL with average sparsity rate as $0.55$, the strongest baseline, FedEM, and the proposed pFedGate with different sparsity rates on EMNIST dataset in Figure 6. We can see that the proposed method achieves comparable convergence speeds with the baselines, which provides empirical evidence for the Theorem 2. In addition, to gain further insights into the proposed method, we plot the histogram of averaged gating weights of all clients on the CIFAR10 dataset in Figure 7. We can see that the sparse pattern becomes stable as we train the gating layer with more local datasets, showing that the proposed method can achieve effective optimization and converge to certain global points under distinct personalized local models.

## H.4 MODEL COMPRESSION MANNER STUDY

We gain sparse local models in the proposed framework via learning personalized gating layers and predicting sparse gated weights for model sub-blocks. As introduced in Section 5.2, we propose a simple and general split manner that chunks the model parameters into $B$ equal-sized sub-blocks, where the split factor $B$ provides flexible splitting granularity. Here we study the sparsification technique proposed in our method by varying the split factor $B$ and comparing it with a SOTA pFL method FedMask Li et al. (2021a), that adopts binary masks to compress the parameters of models' last layers. The results are shown in Table 7, where we examine the FL accuracy by setting $B = [2, 5, 10]$ and $s = [0.3, 0.5]$.

Table 6: Averaged accuracy when randomly sampling 20% clients at each round. The last two columns indicate the average drop compared to non-sampled case.

| | FEMNIST ($\uparrow$) | | CIFAR10 ($\uparrow$) | | Drop ($\downarrow$) | |
| | $\overline{Acc}$ | $\widetilde{Acc}$ | $\overline{Acc}$ | $\widetilde{Acc}$ | $\overline{Acc}$ | $\widetilde{Acc}$ |
|---|---|---|---|---|---|---|
| FedAvg | 77.45 | 57.98 | 66.31 | 57.96 | 3.56 | 2.55 |
| FedEM | 78.54 | 63.37 | 71.12 | 60.76 | 1.45 | 1.78 |
| pFedGate, s=1 | 87.86 | 77.43 | 74.29 | 64.11 | 0.82 | 1.13 |
| pFedGate, s=0.5 | 86.72 | 75.12 | 74.25 | 64.36 | 0.79 | 0.21 |
| pFedGate, s=0.3 | 85.78 | 74.89 | 73.91 | 64.38 | 0.36 | 0.80 |

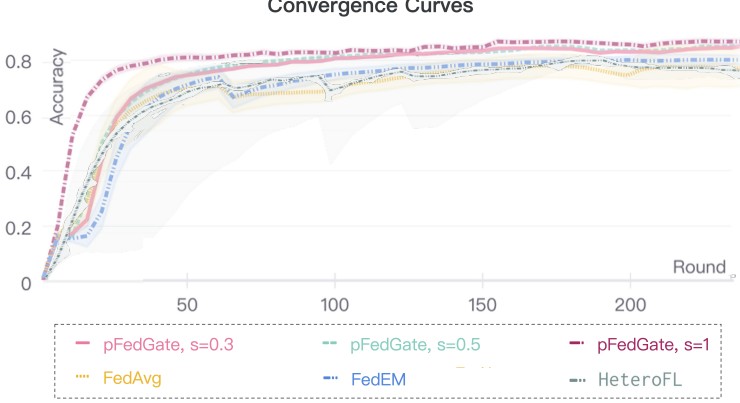

Figure 6: The convergence curves of FedAvg, HereroFL, pFedGate and the strongest baseline, FedEM on EMNIST. The pFedGate achieves comparable or even better ($s$=1) convergence speeds than baselines.

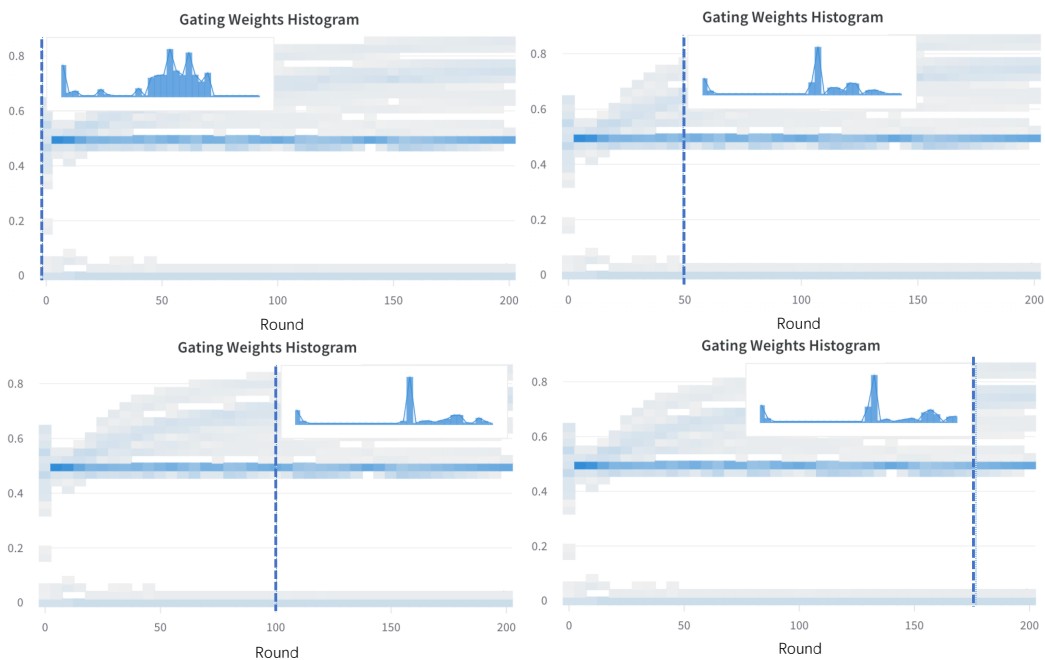

Figure 7: The convergence curves of pFedGate in terms of the histogram of averaged gating weights over all clients on the CIFAR10 dataset.

From Table 7 we have the following observations: Our method achieves better performance than FedMask in various sparsity settings. Besides, when increasing the sparsity ($s = 0.5$ to $s = 0.3$), our method with $B = 5$ still achieves good performance while FedMask gains larger performance drops. These results verify the effectiveness of the proposed adaptive sparse weights prediction again. As for the impact of $B$, on the one hand, we can see that too small $B = 2$ gains relatively worse results than $B = 5$ as there may be insufficient sub-blocks to model the clients' diversity. On the other hand, the larger $B = 10$ achieves similar results to $B = 5$ while larger performance drop when increasing sparsity. We empirically found that $B = 5$ has good robustness and adopt it as the default value in our experiments.

Table 7: Averaged accuracy comparison for pFL methods with and without model compression. The numbers in the parentheses indicate the accuracy difference between sparsity $s = 0.5$ and $s = 0.3$ for FedMask and pFedGate.

| | | | FEMNIST | | CIFAR10 | |
|---|---|---|---|---|---|---|
| | | | $\overline{Acc}$ | $\breve{Acc}$ | $\overline{Acc}$ | $\breve{Acc}$ |
| FedAvg | | | 76.51 | 60.82 | 68.33 | 60.22 |
| FedEM | | | 80.12 | 64.81 | 72.43 | 62.88 |
| FedMask | $s$=0.5 | | 78.69 | 62.18 | 70.54 | 61.37 |
| | $s$=0.3 | | 77.41 ($-1.28$) | 60.69 ($-1.49$) | 68.46 ($-2.08$) | 59.05 ($-1.32$) |
| pFedGate | $B$=5 | $s$=0.5 | 86.31 | 75.68 | 74.07 | 64.21 |
| | | $s$=0.3 | 86.75 ($+0.44$) | 76.47 ($+0.79$) | 73.65 ($-0.42$) | 64.39 ($+0.18$) |
| | $B$=2 | $s$=0.5 | 81.10 | 71.28 | 70.81 | 61.03 |
| | | $s$=0.3 | 80.03 ($-1.07$) | 70.72 ($-0.56$) | 70.65 ($-0.16$) | 60.66 ($-0.37$) |
| | $B$=10 | $s$=0.5 | 85.38 | 74.91 | 72.94 | 63.59 |
| | | $s$=0.3 | 84.67 ($-0.71$) | 75.10 ($+0.19$) | 72.48 ($-0.46$) | 63.58 ($-0.11$) |

