# OpenReview forum: "Efficient Personalized Federated Learning via Sparse Model-Adaptation"
_ICLR.cc/2023/Conference — Submitted to ICLR 2023_

### Official Review · Reviewer_wTvc · 2022-10-23

**Confidence:** 4
**Correctness:** 3
**Technical Novelty And Significance:** 3
**Empirical Novelty And Significance:** 3
**Recommendation:** 6

**Clarity, Quality, Novelty And Reproducibility:**

Clarity:

The paper is very well-written in general. As I commended the authors above, their appendix is very well-structured. I do, however, have a few nitpicks:

1. The presentation of the proving techniques is too terse. It would always be good to start with a narrative of a general proving strategy, followed by step-by-step derivation annotated with explanation why each step holds
2. Section 5.2.3 is very much lacking in detail, which actually raises concerns about the differentiability of the entire workflow -- I will comment more on this below.

Novelty & Quality:

As stated above, problem formulation & solution idea are new so I acknowledge the novelty here. But, I am not sure I follow why the last key step in combining output of the two flows (mask generator & importance score generator)  via knapsack (Section 5.2.3) is differentiable.

I agree that using the previous technique of binarized neural net, the authors can make the representation of the "knapsack matrix" I differentiable. But, it seem the max operator is still not differentiable & so if the solution to Eq. (6) is to be integrated into the "end-to-end" pipeline, it would come out in the form of a bi-level optimizable task -- further treatment of this probably needs to be detailed because otherwise, it is not immediately clear to me how gradient flows through that bi-level structure.

As for the empirical experiment, please do consider comparing with Fallah et al. 2020 too for thoroughness. In addition, there should be key discussions & demonstrations of how this proposed mechanism has an advantage over simpler engineering solution for FL with knowledge distillation. For instance, clients can download big model from the server but will distill into smaller local model with heterogeneous structure before baking in local data; in turn, the server can pull local models from clients and distill soft labels from them to train its global model. How would the proposed model perform against this vanilla approach?

More suggestions:

Although the idea of using heterogeneous models in personalized FL is quite new; it has been explored before (in a different context with a different probabilistic formulation) in standard FL setting. For example,

Bayesian Nonparametric Federated Learning of Neural Network (ICML-19)
Statistical Model Aggregation via Parameter Matching (NeurIPS-19)

Both approaches in these works would allow local models to be heterogeneous in size so as to accommodate for the heterogeneity in their hardware capacity. The formulation is different but it addresses the same aspect so some discussions regarding pros/cons between the two lines of approaches would be nice to have.

More nitpicks:

It would be good to repeat all experiments a few times to generate error bars
Including in the appendix more detail about the differentiable parameterization detailed in (Hubara et al. 2016)
Discussing potential directions to expand the proving technique beyond the strong convexity assumption which is tied to simple linear model

Reproducibility:

The code is released and the algorithm description is sufficiently clear so I believe reproducibility is possible. But still, for a thorough understanding of the approach, Section 5.2.3 needs a significant flesh-out.


**Strength And Weaknesses:**

Strengths:

+ Both problem formulation & proposed solution are novel to me. I have not seen similar approaches to this before.
+ The technical presentation is very well-organized, particularly the well-structured appendix that makes it easy to zoom into each component of the proposed work
+ The empirical results are reasonably extensive with comparison to multiple baselines in the same direction
+ There are interesting theoretical results showing both convergence of the proposed algorithm and the existence of optimal local solution: among optimal local solution that meets local sparse constraints, there exists one that can be acquired via masking the global model

Weaknesses:

- Lacking comparison with a known baseline -- Fallah et al., 2020
- Lacking discussion with other related works on FL with heterogeneous models
- Theoretical results are established under strong convexity of the loss function, which is likely not the case unless the model is linear
- Lacking experiments showing advantages over simpler knowledge distillation technique for FL with heterogeneous model

I will elaborate on those weaknesses with specific comments & suggestions below.

**Summary Of The Paper:**

This paper introduces a new approach to personalized federated learning via sparse model adaptation. The key idea is to configure local models as subnets of a master net representing the global model.

The subnets are generated via a block-wise binary mask that (1) zeroes out less important components; and (2) enforce a certain sparsity level such that the no. of non-zero parameters of resulting subnets fits with local hardware constraints.

Local models are therefore heterogeneous compression of the same global model. This importantly allows local models to have heterogeneous architectures, which is, to my understanding, a less studied aspect of personalized FL. Most existing personalized FL approaches only consider data heterogeneity (but not model heterogeneity).

The generative mechanism of the subset is parameterized by a two-flow model: (1) one flow generates crude masks (that might violate local sparsity constraints); (2) the other flow generates an importance score for each parameter blocks. The output of two flows are combined via solving a knapsack problem, resulting in a binary mask that meets the hard sparsity constraint.

Comparing to existing model compression or gradient quantization techniques, which are often post-training editing approaches, the proposed approach allows learning both the compression mechanism & the updates to the global model.

**Summary Of The Review:**

The paper presents an interesting formulation of personalized FL & a new solution addressing a less studied issue of model heterogeneity. The novelty of this work is strong but its execution needs more polishing, particularly regarding missing experiments with both related work & vanilla baseline (to make sure such proposed mechanism is needed & cannot be bypassed by simpler, trivial engineering solution). The presentation of proving techniques can also be improved. Please refer to my specific comments above. Overall, I believe this will be an acceptable paper if the above are addressed sufficiently.

---

> ### Author Response · Authors · 2022-11-19
> **Responses to Reviewer wTvc, part 2/2**
>
> **[Comparison to mentioned pFL works] 1. "Lacking comparison with a known baseline -- Fallah et al., 2020." 2. "Lacking experiments showing advantages over simpler knowledge distillation technique"**
>
> Thank you very much for mentioning the related pFL works! According to your suggestion, we made the following comparison and add more related discussion in the revision.
> - For the mentioned known baseline, Per-FedAvg [1],
>     - in the "Related Work" Section, we discussed the difference: "it leverages model agnostic meta-learning to enable fast local personalized training that requires computationally expensive second-order gradients". The high computation cost challenges real-world applications, especially with limited system resources.
>     - Besides, in Table 1, we have compared another meta-learning-based method pFedMe [2] that improves Per-FedAvg with faster convergence and smaller computation complexity.
>     - In the response stage, we newly conduct experiments to compare pFedGate with Per-FedAvg. We grid search its learning rates $\alpha$ and $\beta$ from [0.0001, 0.001, 0.01, 0.1, 1, 10]. The results are listed below and we can find that pFedGate achieves both better average bottom accuracy than the known baseline.
>
> - For the related research line of  FL with knowledge distillation (KD),
>     - indeed, the KD technique can help to train heterogeneous architectures in FL and is a promising direction to solve the resource heterogeneous challenge. However, to our best knowledge, many existing KD-based FL/pFL works rely on some common reference object shared among clients, to enable knowledge alignment between the teacher and student models. For example, works [3,4] introduce additional public data for all clients and perform FL with the help of the model logits on the public data. It's worth to notice that such additional common reference may not always available in some scenarios.
>     - Further, we conduct new experiments to compare a known KD-based FL method, FedMD [3]. We list the results on FEMNIST and CIFAR-10 below, where "FedMD, Far-Trans" indicates a far-transfer case by adopting the CIFAR-10 as the public dataset for FEMNIST experiment, and adopting the FEMNIST as the public dataset for CIFAR-10 experiment. The "FedMD, Near-Trans" indicates a near-transfer case by adopting the EMNIST as the public dataset for FEMNIST experiment, and adopting the CIFAR-100 as the public dataset for CIFAR-10 experiment. We find that pFedGate still shows superiority in terms of the average/bottom accuracy, and robustness when we vary the sparsity $s$ for pFedGate and vary the public dataset for FedMD.
>
> [1] Personalized Federated Learning with Theoretical Guarantees, NeurIPS 2020
>
> [2] Personalized Federated Learning with Moreau Envelopes, NeurIPS 2020.
>
> [3] FedMD: Heterogeneous Federated Learning via Model Distillation, NeurIPS 2019
>
> [4] Data-Free Knowledge Distillation for Heterogeneous Federated Learning, ICML 2021
>
> |                 |      FEMNIST    |     |      CIFAR-10 |       |
> |-----------------|---------|------------|---------|------------|
> |                 | Acc-avg | Acc-bottom | Acc-avg | Acc-bottom |
> | FedAvg          |  76.51  |    60.82   |  68.33  |    60.22   |
> | Per-FedAvg      |  77.29  |    61.42   |  70.35  |    60.96   |
> | FedMD, Far-Trans|  61.25  |    51.37   |  55.28  |    52.04   |
> | FedMD, Near-Trans| 75.87  |    60.29   |  67.52  |    59.43   |
> | FedEM           |  80.12  |    64.81   |  72.43  |    62.88   |
> | pFedGate, s=1   |  87.32  |    77.14   |  75.18  |    66.67   |
> | pFedGate, s=0.5 |  86.31  |    75.68   |  74.07  |    64.21   |
> | pFedGate, s=0.3 |  86.76  |    76.47   |  73.65  |    64.39   |
>
>
> **[Discussion on the mentioned FL works] "The formulation is different but it addresses the same aspect so some discussions ... between the two lines of approaches would be nice to have."**
>
> Many thanks for bring the two related works [1,2] to our attention! We have made the following comparison and add related discussion in the revision.
> We agree that the Bayesian FL is related and promising to address the heterogeneous challenge, due to its advantages of nonparametric and probabilistic modeling. This line of works brings a large range of applications in terms of the model and task types, and the ability to flexibly incorporating model parameter priors.
> Meanwhile, in the proposed work, we study a relatively orthogonal sparsification training method, which is based on batch/example level model-adaption and can achieve good accuracy.
>
> Combining these two ideas are interesting, e.g., we may explore the sparsity with Gaussian prior on the local models. Besides, as we shown in the personalization study experiment (Sec.7.3 and Figure 4), we implicitly learn group-wise personalization with the sparsity constraint. We can further improve this with hierarchical probability modeling such as Dirichlet distribution. We leave these as future works and thanks again!

---

> ### Author Response · Authors · 2022-11-19
> **Responses to Reviewer wTvc, part 1/2**
>
>
> Many thanks for the insightful and helpful comments by the reviewer! We make the following point-by-point responses to your comments:
>
> **[Clarity] "The presentation of the proving techniques is too terse. It would always be good to start with a narrative of a general proving strategy, followed by step-by-step derivation annotated with explanation why each step holds."**
>
> Thanks for your helpful suggestion! In the revision, we have improved our proofs for both generalization and convergence in Appendix C and Appendix D, and added more explanations for equations (14-16).
> Specifically, in equation 14 and 16, we give hints of derivation in the important steps, and we also give more descriptions about the notations.
>
>
> **[Clarity] "Section 5.2.3 is very much lacking in detail, which actually raises concerns about the differentiability of the entire workflow"**
>
> Sorry for the confusion. The reason why the combination of these two flows (red G and blue M in Figure 2) is still differentiable is that we used the straight-through trick. Recall that the combination is $M'=M \cdot I^\*$ where $I^\*$ is a binary index and a solution to the knapsack problem.
> The "straight-through" (ST) here means that we build such a computation graph, in which we only use binary variable $I^\*$ in the forward pass stage, while *replace $I^\*$ into the differentiable variable $G$ in backward gradient propagation*.
>
> Specifically, this can be implemented by the following demonstrative PyTorch code:
> `$I^*_{ST}$ = $I^*$ - G.detach() + G`.
> In this way, the tensor $I^*_{ST}$ and $M$ are both differentiable and can be used to pass the gradient flow via their combination tensor $M'=I^*_{ST} * M$ and the following adapted tensor of the sparse model $\theta^{'}_{i} = \theta_g^{t-1} \circledast M'_i$. And thus we can train the personalized model and the gating layer as Algorithm 1, lines 6 and 7 shown.
> Thanks again for the question and comments! We have added more detailed explanations in the revision.
>
>
> **[Assumption] Theoretical results are established under strong convexity of the loss function**
>
> Thank you for the comments!
> We would like to point out that most SOTA FL works and pFL works with convergence proof are also based on the strong convexity such as in [1,2,3,4].
> Relaxing this convexity assumption to make a novel convergence proof may be a great enough contribution and progress, which can inspire and help to extend the theoretical analyses of a great deal of existing FL and pFL works.
> We just have the preliminary idea of analyzing the convergence of the proposed method via neural tangent kernel (NTK) [5]. The training dynamics of pFedGate may could be modeled into some covariance matrices, and the least eigenvalue of the induced kernel can be used to depict the convergence. We leave it as future work.
>
> [1] Towards Personalized Federated Learning. TNNLS, 2022.
>
> [2] Advances and Open Problems in Federated Learning. Foundations and Trends in Machine Learning, 2021.
>
> [3] On the Convergence of FedAvg on Non-IID Data. ICLR, 2020.
>
> [4] On the Convergence of Clustered Federated Learning. arXiv, 2022.
>
> [5] Neural tangent kernel: Convergence and generalization in neural networks. NeurIPS, 2018

---

### Official Review · Reviewer_PyZd · 2022-10-25

**Confidence:** 4
**Correctness:** 2
**Technical Novelty And Significance:** 2
**Empirical Novelty And Significance:** 2
**Recommendation:** 3

**Clarity, Quality, Novelty And Reproducibility:**

The writing is clear, but the proposed method does not live up to the expectation of handling resource-heterogeneous clients. Some of the implementation details are missing.

**Strength And Weaknesses:**

Strengths
- Writing is clear and easy to understand
- Thorough theoretical analysis

Weaknesses
- The framework fails to solve the resource heterogeneity issue since the clients still need to store the whole (global) model.
- Some implementation and experiment details remain unclear.

Detail comments:

1. Sec.4 first paragraph: the two-step process “still requires computational and communication costs corresponding to the un-compressed models during the FL process”. However, according to Fig.1, pFedGate also needs to store the un-compressed model locally in order to compute the compressed model using M’, which is also mentioned in Sec.5.1 “all clients share the same global model \theta_g that downloaded from server”. In addition, since every example uses different blocks of the model, each client has to store the whole model for inference. Therefore, both training and inference are dense, making the main contribution unsubstantiated. (Appendix E is also misleading.)

2. The gradient computation requires further discussion. The forward pass of every example requires solving a knapsack problem as shown in Sec.5.2.3. Combing with the fact that we are training multiple networks (\theta and \phi), it would be helpful to elaborate on how the gradient of each network is computed using the knapsack solution.

3. What is the relative sparsity of gating layer s_{\phi_i} in Sec.6.3?

4. For the experiments
- How many clients are used for the experiments in Sec.7.2?
- Sec.5.2.2 discusses that different DNN layers can have different block structures. Then how are the layers being partitioned in the experiments?

**Summary Of The Paper:**

This paper proposes a personalized federated learning framework that selects a subset of model parameters (thus sparse) for each training example via personalized gating units for each client (Fig.1). The proposed method, pFedGate, has a gating unit/layer for each client, consisting of a rescaling factor M, an importance factor G and learns the binary mask by solving a knapsack problem per training example (Fig.2). Theoretical analysis on generalization, convergence and complexities are provided. Experiments on common personalized FL benchmark datasets show that pFedGate can outperform existing methods.

**Summary Of The Review:**

The paper proposes a framework for personalized FL and it can work well in the experiments. However, it does not provide sufficient justification for the main claim/contribution of solving the resource heterogeneity issue as all clients still need to store the global model locally for training and inference.

---

> ### Author Response · Authors · 2022-11-19
> **Responses to Reviewer PyZd, part 2/2**
>
>
> **[W2.3, question on Sec.6.3] "What is the relative sparsity of gating layer s_{\phi_i} in Sec.6.3?"**
>
> Thanks for the question! The relative sparsity of gating layer $s_{\phi_i}=count(\phi_i \neq 0)/d$, and we have added the clarification in the revision.
>
> **[W2.4.1, question on Sec.7.2] "How many clients are used for the experiments in Sec.7.2?"**
>
> Thank you for the question! In Section 7.1 we mentioned "Due to limited space, please refer to Appendix G for more details about the datasets, models and baselines". And in the Appendix F, we introduce the number of clients, 100, 539, 100, 50 for the adopted EMNIST, FEMNIST, CIFAR10, and CIFAR100 respectively.
>
> **[W2.4.2, question on Sec.5.2.2] "how are the layers being partitioned in the experiments?"**
>
> Thanks for your question! As we mentioned in Section 5.2.2, "we propose a general operator-type-free splitting manner by default", which means that  unless otherwise stated, we adopt the proposed splitting manner with $B$ blocks in the experiments.
> In Section 7.5, we mentioned that "In Appx.G.4, we vary the block splitting
> factor B of pFedGate", and we empirically found that $B=5$ has a good robustness and adopt it as the default value in our experiments.
> Thanks again!

---

> > ### Comment · Reviewer_PyZd · 2022-11-23
> > **Storage Heterogeneity is Ignored**
> >
> > I would like to thank the authors for the additional clarifications. Some of my questions have been addressed. However, my main concern remains. Firstly, the algorithm would not be complete if the heterogeneity in storage/memory is not addressed. By ignoring storage constraints, the paper studies a very narrow learning scenario which is rarely encountered in practice. Secondly, the discussion about the time complexity for communication at each round (Sec.6.3) is misleading. Recall that the server still needs to send the whole model to each client even though the client can return partial models. As a result, the communication is still O(d) per round for each participating client.
> >
> > Overall, the target learning scenario is not very clear for the current paper, and the improvements in computation and communication are not very significant.

---

> > > ### Author Response · Authors · 2022-11-28
> > > **More discussion regarding storage improvement and target scenario**
> > >
> > > Thank you for your response!
> > > - We would like to provide some updated results and discussion regarding storage performance. Indeed, our approach requires storing the dense original model for subsequent sparse adaptation, which brings us with at least 1x the storage overhead relative to the original model. However, note that **FL requires training on client-side**, in addition to the model parameters，the memory cost in FL training process also contains the **intermediate tensors in both the forward and the backward propagation phases**, whose size depends on the model architecture, the shape of the inputs, and the specific implementation. Thus once the model-adaption of pFedGate is completed, *the intermediate tensor involved in the forward and backward processes through the sparse model is correspondingly smaller based on block sparsity*, allowing us to get a lower peak training memory than FedAvg. We have tracked the peak memory in Mb for one client and one FL round, and list the results on the CIFAR-10 dataset with different batch size below. We can see that the proposed method indeed improved the memory cost. For example, when $s$=0.3 and $bz=128$, pFedGate uses only 66% peak memory compared with FedAvg, meanwhile achieving 5.32% average accuracy and 4.17% absolute improvements (Table 1. Sec.7.2).
> > >
> > > - Recall that the additional storage cost introduced by the gating layer is very light-weight, whose shape is about the multiple of dimension of the input feature and the number of the total sub-blocks. E.g., the size of gating layer $d_{\phi_i}$ is only 6.3% of the original model for the CNN model used on CIFAR-10. **We would like to point out that pFedGate actually has great potential for further storage savings for multi-task applications.** Considering that if we use one dense original model but learn $T$ distinct gating layers for different $T$ downstream tasks on a client, the storage cost will be $d+T*d_{\phi_i}$. We will explore to extend the current sample/batch-level adaptation to task-level adaptation in the future.
> > >
> > > - Finally, we agree that the communication cost is currently reduced only in the client upload phase. However, client devices such as mobile phones usually have smaller upload bandwidth than download due to the asymmetric internet connections, and server usually has stronger network resources and more flexible strategies (e.g. small fraction of clients sampled, asynchronous FL, etc.) to handle the model broadcast process.
> > > With these new discussions in mind, we would like to clarify that the proposed method aims to practical learning scenarios, and does improve accuracy and system efficiency at the same time as the experiments shown.
> > >
> > > We sincerely appreciate that the reviewer consider increasing your overall score of the paper if our responses convince you of our work's contribution.  Please feel free to let us know if you have any additional questions or suggestions. Thanks again!
> > >
> > >
> > > | Peak-Mem (Mb) | bz = 32 | bz = 64 | bz = 128 | bz = 256 | bz = 512 |
> > > |--------|---------|---------|----------|----------|----------|
> > > | FedAvg | 26      | 26.3    | 27.1     | 28.6     | 31.8     |
> > > | pFedGate, s=1    | 26.8    | 27.2    | 27.9     | 29.4     | 32.4     |
> > > | pFedGate, s=0.5  | 21      | 21.5    | 22.2     | 23.7     | 26.7     |
> > > | pFedGate, s=0.3  | 16.9    | 17.3    | 18       | 19.7     | 22.9     |
> > > | pFedGate, s=0.1  | 14.9    | 15.2    | 16.1     | 17.5     | 20.5     |
> > >
> > > *bz denotes batch size.

---

> ### Author Response · Authors · 2022-11-19
> **Responses to Reviewer PyZd, part 1/2**
>
>
> Thank you very much for your insightful comments! We make the following responses point by point to address your comments:
>
> **[W1 & W2.1] "The framework fails to solve the resource heterogeneity issue since the clients still need to store the whole (global) model." "pFedGate still requires computational and communication costs corresponding to the un-compressed models during the FL process"**
>
> Many thanks for your comment!
> - **[Storage cost]**
> We would like to clarify that the resource heterogeneity issue involves several aspects in terms of the devices' hardware capacities, such as computation, communication and storage. In this paper, we focus on reducing **computation and communication cost** with a tolerable storage cost. To make this more precise and clear, we have modified our related claims in the revision, to emphasize that pFedGate mainly focuses on computation and communication efficiency.
> As for the storage cost, we agree that pFedGate still requires to store the whole model in each client, and we will improve it in future work.
> However, we argue that compared with FedAvg, pFedGate only introduces a  light-weight adaptive gating layer, which is negligible compared with numerous existing personalization solutions that need to store both global model and at least one personalized local model object [1,2,3].
>
> - **[Computation and communication costs]**
> Meanwhile, we emphasize that pFedGate has significant gain in both computation and communication.
> Regarding computation cost, since pFedGate will **remove some blocks** and generate **a sparse sub-network**, it actually reduces the computation cost in both inference and training even without sparse matrix computing libraries.
> As for communication, though pFedGate uses different blocks for different batches (or samples when batch size is 1), similar with most network pruning techniques [4], the knapsack modular in pFedGate usually learns to reserve the more important blocks and weights.
> As a result, only partial weights are changed during local training, and finally uploaded to the server.
> To further prove the effectiveness of pFedGate in communication, in Appendix E we have conducted experiments and we observe that only 67% weights are uploaded on average, when we run FL experiments 3 times on FEMNIST and CIFAR10 with $s_i$ = 0.5, local update step as 1 epoch, and batch size as 128.
> We also provide more discussion about the average non-zero parameter ratio $q_i$ in Appendix E. Thanks again!
>
> [1] Ditto: Fair and robust federated learning through personalization. ICML, 2021.
>
> [2] Federated multi-task learning under a mixture of distributions. NeurIPS, 2021.
>
> [3] Towards Personalized Federated Learning. TNNLS, 2022.
>
> [4] Pruning and Quantization for Deep Neural Network Acceleration: A Survey. Neurocomputing, 2021.
>
>
> **[W2.2, Unclear details for gradient computation] "it would be helpful to elaborate on how the gradient of each network is computed using the knapsack solution."**
>
> Sorry for the confusion. The reason why the combination of these two flows (red G and blue M in Figure 2) is still differentiable is that we used the straight-through trick. Recall that the combination is $M'=M \cdot I^\*$ where $I^\*$ is a binary index and a solution to the knapsack problem.
> The "straight-through" (ST) here means that we build such a computation graph, in which we only use binary variable $I^\*$ in the forward pass stage, while *replace $I^\*$ into the differentiable variable $G$ in backward gradient propagation*.
>
> Specifically, this can be implemented by the following demonstrative PyTorch code:
> `$I^*_{ST}$ = $I^*$ - G.detach() + G`.
> In this way, the tensor $I^*_{ST}$ and $M$ are both differentiable and can be used to pass the gradient flow via their combination tensor $M'=I^*_{ST} * M$ and the following adapted tensor of the sparse model $\theta^{'}_{i} = \theta_g^{t-1} \circledast M'_i$. And thus we can train the personalized model and the gating layer as Algorithm 1, lines 6 and 7 shown.
> Thanks again for the question and comments! We have added more detailed explanations in the revision.

---

### Official Review · Reviewer_sfrQ · 2022-10-31

**Confidence:** 4
**Correctness:** 3
**Technical Novelty And Significance:** 3
**Empirical Novelty And Significance:** 4
**Recommendation:** 5

**Clarity, Quality, Novelty And Reproducibility:**

The paper has typos, but otherwise is well written. It also have novelty in the method proposed. I have not verified all the proofs or tried to run the provided code.

**Strength And Weaknesses:**

Strengths:
1. The proposed framework seems to achieve better personalization accuracy than existing models.
2. The idea of generating personalized masks (gating weights) using a computationally cheap network is novel.


Concerns / Questions:
1. I think Assumption 1 might be a strong assumption, since it assumes bounded diversity between the globally optimal model, and the optimal sparse local model. In particular, I think $\sigma_i$ might itself depend on the level of sparsity for the device.

2. In the proof of proposition 1, I do not follow how the Reverse Cauchy-Schwarz inequality is applied, since the left hand side of Eq. (13) has the l2 norm squared of element-wise multiplication of vectors instead of inner product. To be precise, the left hand side will look like $(\sum_j (M^*_{i,j}\theta^*_{g,j})^2 )$ instead of $(\sum_j (M^*_{i,j}\theta^*_{g,j}) )$. Also, I think the denominator of Eq. (4) should not have the square in norm of $\theta^*_{g}$.

3. The experimental results do not provide the time (in seconds) and memory requirements (in GB / MB) for the training of the models. Further, the convergence vs communication round curve is only provided for EMNIST dataset. This data is important since one of the motivations for this paper is to perform Federated Learning efficiently on computationally slow devices.

4. In Eq. (7), is the gradient w.r.t $\phi$ as well? In other words, does Theorem 2 prove convergence for the gating layer as well?

5. On page 6, section 6.1, I think the definition of $L(\theta_g, \phi_i)$ and $\hat{L}(\theta_g, \phi_i)$ should not have the $i$ in subscript on the left hand side since it also appears in the summation on the right.

**Summary Of The Paper:**

The paper proposes a new framework called pFedGate for training personalized models at client devices, such that each model can have different sparsity level depending on the devices computational and memory capacities. The idea is to generate personalized masks with different sparsity levels for each device, using a small gating layer per device. The gating layer itself is small in size and hence the paper claims that it does not add much overhead. The empirical results show that this technique indeed achieves better personalization than existing methods.

**Summary Of The Review:**

While the proposed idea of generating personalized masks (gating weights) is novel and has potential, I have some concerns regarding the theory and experimental results that I have mentioned above.

---

> ### Author Response · Authors · 2022-11-19
> **Responses to Reviewer sfrQ**
>
> Many thanks for your careful review and helpful comments! Due to time constraints, we have updated the revision first, and we are still working on a detailed response to your comments. We will post the full point-by-point content as soon as possible.

---

> ### Author Response · Authors · 2022-11-28
> **Responses to Reviewer sfrQ, part 2/2.**
>
>
> **[C/Q3] Experimental results w.r.t training time, memory and convergence rounds**
>
> - Thank you very much for your helpful suggestions! We have tracked the training time in ms and peak memory in Mb for one client during one FL round. The results on the CIFAR-10 dataset with different batch sizes are listed below. We can see that the proposed method indeed improved the training efficiency and memory cost, where the peak memory is in Mb and the training time is in ms and averaged by the number of trained data batches. For example, when $s$=0.3 and $bz=128$, pFedGate uses 66% peak memory and 72% training time compared with FedAvg, meanwhile having 5.32% average accuracy and 4.17% absolute improvements (Table 1. Sec.7.2).
>
> - It is worth to mention that in addition to the model parameters, the training process (including the backward propagation phase) also requires storing the intermediate tensors, whose size depends on the model architecture, the shape of the inputs, and the specific implementation. Thus although pFedGate requires storing the full amount of the original dense model, once the model-adaption is completed, the intermediate tensor involved in the forward and backward processes through the sparse model is correspondingly smaller based on block sparsity, allowing us to get a lower training memory peak and less training time than FedAvg.
>
> |  | bz = 32 |  | bz = 64 |  | bz = 128 |  | bz = 256 |  | bz = 512 |  |
> |---|---|---|---|---|---|---|---|---|---|---|
> |  | Peak-Mem | Train-Time | Peak-Mem | Train-Time | Peak-Mem | Train-Time | Peak-Mem | Train-Time | Peak-Mem | Train-Time |
> | FedAvg | 26 | 3.07 | 26.3 | 3.25 | 27.1 | 3.89 | 28.6 | 4.23 | 31.8 | 4.51 |
> | pFedGate, s=1 | 26.8 | 3.65 | 27.2 | 3.04 | 27.9 | 4.09 | 29.4 | 4.29 | 32.4 | 4.66 |
> | pFedGate, s=0.5 | 21 | 2.44 | 21.5 | 2.58 | 22.2 | 3.21 | 23.7 | 3.39 | 26.7 | 3.58 |
> | pFedGate, s=0.3 | 16.9 | 2.32 | 17.3 | 2.42 | 18 | 2.81 | 19.7 | 3.07 | 22.9 | 3.33 |
> | pFedGate, s=0.1 | 14.9 | 2.28 | 15.2 | 2.31 | 16.1 | 2.68 | 17.5 | 2.85 | 20.5 | 3.09 |
>
> - As for the convergence rounds, we present the results on all four adopted datasets in terms of the FL round that each method achieves the 90% of its highest average accuracy in the below table, where the values are averaged over three experiments with different random seeds. We observe that pFedGate gains comparable convergence speed with FedAvg, providing empirical evidence for Theorem 2. With the results and discussion of the single-FL-round and single-client efficiency given above, we can see that pFedGate is indeed able to achieve significant accuracy improvements while at the same time improving system efficiency.
>
> |        | EMNIST | FEMNIST | CIFAR-10 | CIFAR-100 |
> |--------|--------|---------|----------|-----------|
> | FedAvg | 72.3   | 195.3   | 253.1    | 226.9     |
> | pFedGate, s=1    | 54.6   | 152.9   | 190.9    | 183.8     |
> | pFedGate, s=0.5  | 57.9   | 174.1   | 219.3    | 196.5     |
> | pFedGate, s=0.3  | 75.2   | 214.4   | 271.8    | 260.7     |
>
>
> **[C/Q4] "In Eq. (7), is the gradient w.r.t $\phi$ as well? In other words, does Theorem 2 prove convergence for the gating layer as well?"**
>
> Thanks for your question! The answer is Yes. As we mentioned in the beginning of the Sec.6.2, "the learnable parameters of pFedGate ($\theta_g$ and {$\phi_i$} where $i \in \mathcal{C}$) converge to stationary points, meanwhile the global model updates become arbitrarily small." This is due to the fact that the sparse model-adaption is differentiable in the proposed method and the gradients flow passes $\theta$ and then $\phi$, as we clarified in Section 5.2.3 with the help of the straight-through trick. In Appendix H.3 (Fig.7), we show empirical evidence for the convergence in terms of the histogram of averaged gating weights.
>
> **[C/Q5] subscript $i$ on page 6.**
>
> Thank you very much to point out this! We mistyped the summation in the right hand of the definitions, where the subscript $i$ in these two losses are actually defined in terms of client $i$. Thus the correct form of the empirical loss is
>
> $\hat{\mathcal{L}}(\theta_g, \phi_i)  \triangleq \frac{1}{|S_i|} \sum_{(x, y) \in S_i} f\Big(\big(\theta_g \circledast \textit{g}_{\phi_i}(x)\big); x, y \Big)$.
>
> And the one for the expected loss is $\mathcal{L}(\theta_g, \phi_i) \triangleq  E_{(x, y) \sim D_i} f\Big(\big(\theta_g \circledast \textit{g}_{\phi_i}(x)\big); x, y \Big)$.
>
> The subscript error here does not affect the proof of Theorem 1, and we'll fix it in the final version. Thanks again for your careful review!
>
> We sincerely thank the reviewer for helping us improve the paper with these accurate comments! Please let us know if our responses have addressed your concerns. We will be ready for further clarification.

---

> ### Author Response · Authors · 2022-11-28
> **Responses to Reviewer sfrQ, part 1/2.**
>
>
> Thank you very much for your careful and helpful review! We make the following point-by-point responses for your comments:
>
> **[C/Q1] Assumption 1 might be a strong assumption.**
>
> Thank you for the comments! The assumption 1 (bounded diversity) is necessary to a feasible and effective FL process, in which there is knowledge that can be shared and mutually beneficial between the FL participants. We would like to clarify that we adopt **a weaker form** than the similar assumption that is also widely used in related literature such as [1,2,3,4].
>
> For example, let's considering the assumption 3 in [2]: "(Bounded diversity). The variance of local gradients to global gradient is bounded as $\frac{1}{N}\sum_{i=1}^N ||\nabla f_i(w) - \nabla_\theta f(w)||^2 \leq \sigma^2_f. $ ". By contrast, our assumption is "There exist a $D_{g}$ such that the variance between local and global gradients is bounded as $||\nabla_\theta f(\theta^*_i) - \nabla_\theta f(\theta^*_g)||^2 \leq \sigma^2_i, ~~~\forall i \in \mathcal{C}$". Note that our assumption is actually **a special case of their form** on sparse model, with our $f(\theta^*_i)$ term corresponding to their $f_i(w)$ term, and our $f(\theta^*_g)$ term corresponding to their $f(w)$ term.
>
> [1] On the Convergence of Clustered Federated Learning. arXiv, 2022.
>
> [2] Personalized Federated Learning with Moreau Envelopes. NeurIPS, 2020.
>
> [3] Federated Multi-task Learning Under a Mixture of Distributions. NeurIPS, 2021.
>
> [4] Think Locally, Act Globally: Federated Learning with Local and Global Representations. NeurIPS, 2019.
>
>
> **[C/Q2] Proof of the proposition 1**
>
> Thanks for your careful review. Yes, there is a mistake when we apply the inverse Cauchy-Schwarz inequality. We correct equation (13) as follows:
>     $A_i ||(M_i-1) \otimes \theta_g^*|| \geq \sqrt{||f\otimes f||||g \otimes g||},$
> where $f:= (M_i^*-1)\otimes (M_i^*-1)$, $g:= \theta_g^* \otimes \theta^*_g$,  $\otimes$ denotes element-wise multiplication.
>
> And $A_i = \frac{1}{4}((R_{\theta_i}^2R_{\theta_g}^2 + r_{\theta_i}^2r_{\theta_g}^2)^2) / (R_{\theta_i}^2R_{\theta_g}^2 r_{\theta_i}^2r_{\theta_g}^2) \leq \frac{1}{4}(R_{\theta_i}^2R_{\theta_g}^2) / (r_{\theta_i}^2r_{\theta_g}^2) $.
>
> Combining Equation (12) and the updated Equation (13) above, we have the updated upper bound of $M_i^*$ as:
> $||f\otimes f||\leq \frac{1}{16}(R_{\theta_i}^4R_{\theta_g}^4 r_{\theta_i}^4r_{\theta_g}^4) / (\sigma_i^4 \mu^4 ||g\otimes g||) $.
>
> We want to thank you again for pointing out this mistake in the derivation of equation (13), and we will update the correction in our final version.
>
> Further, we also want to mention that proposition 1 just show the motivation of the proposed method, and the detailed upper bound of $M^*_i$ is not involved in our remaining theoretical analysis. Therefore, *the updating of the upper bound $M_i^\*$ doesn't affect the remaining theoretical analysis*. Thanks again!

---

### Official Review · Reviewer_gH8r · 2022-11-01

**Confidence:** 5
**Correctness:** 1
**Technical Novelty And Significance:** 2
**Empirical Novelty And Significance:** Not applicable
**Recommendation:** 6

**Clarity, Quality, Novelty And Reproducibility:**

The paper is clearly expressed with good quality.
Novelty needs to be discussed in the two papers not cited.
Reproducibility is uncertain.

**Strength And Weaknesses:**

Strength:

S1). The authors provided a theoretical analysis of the proposed algorithm.

S2). The method is validated with sufficient experiments.

S3). The results are very promising.

Weakness:

W1). The novelty of the paper is not very high since there are already some papers with similar ideas in FL. As claimed in Related work “our method generates different sub-models from the whole global model with a larger learnable parameter space to handle heterogeneous data distribution”, actually the following work is using a similar idea. It also adopts the block-wise selection from the global model and generates personal and sparse architecture for different clients.
a). Personalized Federated Learning via Heterogeneous Modular Networks. IEEE ICDM 2022.
https://arxiv.org/pdf/2210.14830.pdf

b). Personalized Neural Architecture Search for Federated Learning, Minh Hang et al. Neurips Workshop 2021.
https://neurips2021workshopfl.github.io/NFFL-2021/papers/2021/Hoang2021.pdf

Actually, these two papers are using more elegant solutions for PFL. The proposed solution adopts the mixture method of ideas of NAS and FL.


W2). It is strongly suggested to include the two papers above in the revised version for discussion of novelty of the proposed approach. B) is discussing the fine-grained subnetwork selection and A) is discussing block-wise subset network selection for communication efficiency that is similar as this paper.


**Summary Of The Paper:**



The draft proposed an approach for efficient personalized FL by adaptively and efficiently learning sparse local models(a subset of the global one) with a shared global model. With a lightweight trainable gating layer, the proposed algorithm enables clients to reach their full potential in model capacity by generating different sparse models accounting for both the heterogeneous data distributions and resource constraints.

**Summary Of The Review:**

Basically,  The paper is of good quality. Although the novelty is not very high since there are two existing papers that adopt similar ideas for Personalized FL, this paper provides a good theoretical analysis and the results are very promising. If the authors can add a good discussion on the novelty compared with the two papers mentioned below, I would champion the acceptance.

---

> ### Author Response · Authors · 2022-11-19
> **Responses to Reviewer gH8r**
>
>
> **W1 & W2: Novelty and comparison to the two mentioned works.**
>
> Thank you very much for your appreciation and helpful comments!
> We clarify that the proposed solution pFateGate is highly different from the solutions within [1,2] as follows, and add more discussion about the novelty in related work section in the revision.
> - For FedMN [1]:
>   - [**Theoretical Analyses**] Compared with FedMN, the performance of pFedGate is theoretically guaranteed for both generalization and convergence (more details about the guarantees please refer to Theorem 1 and 2 in Section 6). While FedMN doesn't provide any theoretical analyses for generalization or convergence. Considering [1] involves a "routing hypernetwork" in their solution, the convergence is crucial for FedMN.
>   - [**Model Capacities**] Besides, the proposed pFedGate method adopts a much larger hypotheses space than FedMN. In FedMN, each modular is either connected or dropped. While pFedGate doesn't only decide the connections of network (connected or dropped), but also scales weights of the remained blocks simultaneously to improve the personalized performance.
> - For FedPNAS [2]:
>   - [**Problem Setting**] the target and studied problem of FedPNAS and pFedGate are different. FedPNAS assumes that FL participants have sufficient hardware resources, which is often the case in cross-silo setting. Based on this assumption, FedPNAS improves the personalization performance via NAS. Differently, our work pFedGate focuses on the case where the clients' resources, such as computation and communication resources, are usually very limited, which is common in the cross-device setting.
>   - [**Computation Cost**] Indeed, searching fine-grained sub-network with NAS is interesting and intuitive. However, FedPNAS cannot be adopted to our studied problem directly, since the NAS process is highly compute-intensive. The search space of FedPNAS is exponentially large as $\mathcal{O}(|A|w)$ where $|A|=2^40$ indicates the possible architecture candidates and $w$ is the number of model parameters in the scale of $10^6$ in their paper, which raises big challenges for the real-world applications especially with limited clients' resources.
>
> - [**Empirical Comparison**] In addition, we further conduct new experiments to compare pFedGate with FedMN [1] and FedPNAS [2] using their official codes. We adopt the same experimental setting as in Table 1. For FedPNAS, we adopt the default sampler type as "cell_based" and grid search its warm-up epochs from [25, 50, 100]. For FedMN, we grid search the hidden dim of x and hidden dim of y in the policy net from [64, 128, 256], and the pre-train epochs. The results on FEMNIST and CIFAR-10 are listed below, and we can find that pFedGate indeed gains better average accuracy and bottom accuracy than these two new baselines, providing empirical evidence of the superiority of the proposed method again.
>
>
> |                 |      FEMNIST |        |      CIFAR-10  |      |
> |-----------------|---------|------------|---------|------------|
> |                 | Acc-avg | Acc-bottom | Acc-avg | Acc-bottom |
> | FedAvg          |  76.51  |    60.82   |  68.33  |    60.22   |
> | FedPNAS         |  78.24  |    62.35   |  68.91  |    60.73   |
> | FedMN           |  81.07  |    63.29   |  71.13  |    62.54   |
> | FedEM           |  80.12  |    64.81   |  72.43  |    62.88   |
> | pFedGate, s=1   |  87.32  |    77.14   |  75.18  |    66.67   |
> | pFedGate, s=0.5 |  86.31  |    75.68   |  74.07  |    64.21   |
> | pFedGate, s=0.3 |  86.76  |    76.47   |  73.65  |    64.39   |
>
> [1] Personalized Federated Learning via Heterogeneous Modular Networks, ICDM 2022
>
> [2] Personalized Neural Architecture Search for Federated Learning. NeurIPS Workshop

---

> > ### Comment · Reviewer_gH8r · 2022-11-21
> > **Concerns addressed**
> >
> > The authors addressed my concerns on novelty.

---

### Official Review · Reviewer_eJXn · 2022-11-02

**Confidence:** 3
**Correctness:** 3
**Technical Novelty And Significance:** 2
**Empirical Novelty And Significance:** 3
**Recommendation:** 6

**Clarity, Quality, Novelty And Reproducibility:**

**Presentation** : A presentation of this paper appears quite dense, so I’m concerned about violation of the ICLR2023 template.

**Strength And Weaknesses:**

### Strength

- This paper consistently presents empirical performance improvement in various experimental results.
- It also suggests sample-wise local models by using gating function $g_{\phi}$, so that we do not practically need to design sophisticated manipulation for it, since this gating function can be learnt during training. It obviously leads to work in FL that disregard data and resource heterogeneity.
- This paper offers many ablation studies of learned FL models, including ones that distinguish each client without additional training. This empirical studies appears relevant for deriving performance improvement of all cases, and supports author's claims.

### Weakness

- This paper should clarify a limit of data and resource heterogeneity of FL framework. In the manuscript, their voices are likely to be misunderstood, so that the proposed method may be used to totally different devices like edge devices and high-performance cloud computation. According to my understanding, this method supposes that all devices have sufficient memory to simultaneously load global and gating function, but each device's computation capability varies.

- Along with the above comment, the authors should reveal the how extents of sparsity this model can support. Particularly, they should investigate when a gating function rapidly deteriorates. Also, they need to show characteristic of representation capability between global and local models. Lastly, they should clearly reveal a limitation and boundary of this method by changing communication frequency and data distribution of each client.

- In technical aspects,  theoretical reasonings in this paper somewhat mismatch their claim that emphasizes sparsity of each client. Especially, in theorem 1 and 2, they do not consider the sparsity of each client, as well as not provide any high-level understanding of this sparsity based on those theorems. Instead, through very strong assumption like strong convexity of all losses and independence with respect to global, gating functions and these parameters $\phi_i$, it simply reveals that the proposed method does not violate the previous FL framework from point of the number of effective samples and convergence analysis.

- This paper must compare of FedMask[1] in terms of masking characteristics by varying sparsity. The FedMask also employs masking parameters for each client, so it is completely compatible with the proposed method excluding sample-wise models. If the authors want to highlight practical usage of heterogeneity of data and resources, this comparison must be presented for better understanding of this paper. However, in experimental results on the manuscript and appendix, they simply describe the proposed method outperform those algorithms.

**Summary Of The Paper:**

### Summary

This paper addresses data and resource heterogeneity of each client in Federated Learning. This paper proposes a gating function $g_{\phi}(x, s_i)$ that allows easily adaptation to each client while taking sparsity into account like computation resources. As gating function generates masking parameters for local models, it can response to various model-size, heterogeneous dataset and computation resources. Especially, the authors suggest that sample-wise model is better effective than a client-wise local model in personalized FL.

Technically, the authors demonstrate that the proposed method does not hurt the previous theoretical convergence bound and generalization gap in FL. This empirically achieves the state-of-art average performance in personalized FL, ensuring bottom decile performance in a variety of experimental results like EMNIST, FEMNIST, CIFAR10 and CIFAR100.

**Summary Of The Review:**

In summary, I believe the paper holds an interesting research direction worth exploring in personalized FL. However, in my opinion, the current manuscript does not meet the standards for acceptance at ICLR 2023. The authors are strongly encouraged to present the limitations of the method by empirical manners, and improve technical aspects as I do believe this is a promising idea.

-------------------

**After rebuttal** : I'm satisfied with the authors feedback related to experimental results. They provide several experimental evidences to support the efficiency of their model and performance improvement via this rebuttal phase. However, I believe that theoretical aspects must be revised and re-organized in the camera-ready version. Furthermore, since the authors have responded my comments after the end of discuss period, so they were not able to revise the manuscript. Under the condition that author assure all comments will be included in the camera ready version, I score this paper as 6.

---

> ### Author Response · Authors · 2022-11-19
> **Initial Responses to Reviewer eJXn**
>
> Thank you very much for your insightful comments! Due to time constraints, we start with the following response and will add more responses to the remaining comments as soon as possible:
>
> **W1 "This paper should clarify a limit of data and resource heterogeneity of FL framework." "According to my understanding, this method supposes that all devices have sufficient memory, ..., but each device's computation capability varies."**
>
> Many thanks for your suggestion, and your understanding is right!
> - We would like to clarify that the resource heterogeneity issue involves several aspects in terms of the devices' hardware capacities, such as computation, communication and storage. In this paper, we focus on reducing **computation and communication cost** with a tolerable storage cost. *To make this more precise and clear, we have modified our related claims in the revision, to emphasize that pFedGate mainly focuses on computation and communication efficiency. *
> As for the storage cost, we agree that pFedGate still requires to store the whole model in each client, and we will improve it in future work.
> However, we argue that compared with FedAvg, pFedGate only introduces a  light-weight adaptive gating layer, which is negligible compared with numerous existing personalization solutions that need to store both global model and at least one personalized local model object [1,2,3].
>
> - Meanwhile, we emphasize that pFedGate has significant gain in both computation and communication. Since pFedGate will **remove some blocks** and generate **a sparse sub-network**, it actually reduces the computation cost in both inference and training even without sparse matrix computing libraries. And in the communication stage, we do not need to transmit the zero sub-blocks. The server still can aggregate the sparse model parameters according to the key-value pairs for the block index and block parameters. Thanks again!
>
> [1] Ditto: Fair and robust federated learning through personalization. ICML, 2021.
>
> [2] Federated multi-task learning under a mixture of distributions. NeurIPS, 2021.
>
> [3] Towards Personalized Federated Learning. TNNLS, 2022.

---

> > ### Comment · Reviewer_eJXn · 2022-11-21
> > **Response to authors comment**
> >
> > Thank you for your response. As you mentioned, the authors only covered the efficiency of the proposed algorithm compared of previous works. However, my main concerns still remain. I'm waiting for next responses, and then I will re-evaluate this paper.

---

> ### Author Response · Authors · 2022-11-28
> **Responses to Reviewer eJXn, part 5/5.**
>
>
> **[Further clarification for the storage]**
>
> In addition to the initial response regarding the storage cost of pFedGate, we now would like to provide some updated results and discussions.
>
> 1. - Indeed, our approach requires storing the dense original model for subsequent sparse adaptation, which brings us with at least 1x the storage overhead relative to the original model. However, note that **FL requires training on client-side**, in addition to the model parameters, the memory cost in FL training process also contains the **intermediate tensors in both the forward and the backward propagation phases**, whose size depends on the model architecture, the shape of the inputs, and the specific implementation.
>
>    - Thus once the model-adaption of pFedGate is completed, *the intermediate tensor involved in the forward and backward processes through the sparse model is correspondingly smaller based on block sparsity*, allowing us to get a lower peak training memory than FedAvg. We have tracked the peak memory in Mb for one client and one FL round, and list the results on the CIFAR-10 dataset with different batch size below. We can see that the proposed method indeed improved the memory cost. For example, when $s$=0.3 and $bz=128$, pFedGate uses only 66% peak memory compared with FedAvg, meanwhile achieving 5.32% average accuracy and 4.17% absolute improvements (Table 1. Sec.7.2).
>
> 2. Besides, Recall that the additional model storage cost introduced by the gating layer is very light-weight, whose shape is about the multiplication of the dimension of the input feature and the number of the total sub-blocks. E.g., the size of gating layer $d_{\phi_i}$ is only 6.3% of the original model for the CNN model used on CIFAR-10. **We would like to point out that pFedGate actually has great potential for further storage savings for multi-task applications.** Considering that if we use one dense original model but learn $T$ distinct gating layers for different $T$ downstream tasks on a client, the storage cost will be $d+T*d_{\phi_i}$. We will explore to extend the current sample/batch-level adaptation to task-level adaptation in the future.
>
> Finally, we acknowledge the very positive impact your comments had on our work. We also hope these responses can address your concerns and convince you to lean more toward acceptance of the paper. Thanks again!
>
> | Peak-Mem (Mb) | bz = 32 | bz = 64 | bz = 128 | bz = 256 | bz = 512 |
> |--------|---------|---------|----------|----------|----------|
> | FedAvg | 26      | 26.3    | 27.1     | 28.6     | 31.8     |
> | pFedGate, s=1    | 26.8    | 27.2    | 27.9     | 29.4     | 32.4     |
> | pFedGate, s=0.5  | 21      | 21.5    | 22.2     | 23.7     | 26.7     |
> | pFedGate, s=0.3  | 16.9    | 17.3    | 18       | 19.7     | 22.9     |
> | pFedGate, s=0.1  | 14.9    | 15.2    | 16.1     | 17.5     | 20.5     |
>
> *bz denotes batch size

---

> > ### Comment · Reviewer_eJXn · 2022-11-29
> > **Response to authors comment**
> >
> > I appreciate authors effort in clarifying all questions along the process, as well as in improving the manuscript. At this point, I'm happy to see improvement of this paper from empirical prospective. Particularly, I have no further questions except Theorem 2. Nonetheless, I'm still wondering that theorem 2 is useful to support this paper. If the authors think about that It does not contribute to this paper, they should remove this point for a clear understanding. During this rebuttal period, the most of issues are resolved, but theoretical aspect is still blurry. In this regard, I raise my score to 6.

---

> > > ### Author Response · Authors · 2022-11-30
> > > **Thank you for your reply and recognition**
> > >
> > > Thank you so much for the quick reply, the recognition of our efforts, and the raising in the overall score! As we are no longer able to upload revisions, we promise to fully reflect these helpful comments and discussions in the final version.
> > >
> > > Regarding Theorem 2, we would like to clarify more about its contribution to this paper in terms of the following two reasons:
> > > - [View of guarantee] Note that in addition to the base model in the typical FL setting, our work *introduces a lightweight auxiliary gating model for sparsification*. In fact, using a naive sparsification technique may harm the loss convergence [1,2], e.g., [1] found that directly optimizing $l_p$ ($p \in \{1,2\}$) sparsity-promoting regularization makes the gradient computation involve not only the original training loss, but also the non-smooth sparsity regularizer, leading to non-convergence (see the loss curves and analysis in their Appendix D). With Theorem 2, we provide a **guarantee** of the convergence of both the base and **gating models**.
> > >
> > > - [View of efficiency improvement] More importantly, recall that the total amount of system overhead in the whole FL process is approximately a product of the single-FL-round overhead and the number of FL convergence rounds. In terms of convergence: our theoretical results about Theorem 2 show that pFedGate has *the same order of the rate of convergence* as a large class of existing work such as FedAvg and FedEM. In terms of single-FL-round overhead: the proposed method reduces communication and computation costs in the single FL round as we showed in the paper and the responses. Overall, our proposed method leads to a **reduction in system overhead across the whole FL process**, since it **does not sacrifice convergence speed** and reduces communication and computation costs in the single FL round.
> > >
> > > [1] Reweighted Proximal Pruning for Large-Scale Language Representation. ArXiv, 2019
> > >
> > > [2] Slowness and sparseness have diverging effects on complex cell learning. PLoS computational biology, 2014.

---

> ### Author Response · Authors · 2022-11-28
> **Responses to Reviewer eJXn, part 4/5.**
>
>
> **[W3.2 sparsity in theorems 2] "in theorem 1 and 2, they do not consider the sparsity of each client, as well as not provide any high-level understanding of this sparsity based on those theorems"**
>
> For theorem 2, we would like to note that in the convergence analysis, we considered the introduced gating layer, which generates the sparsity. With theorem 2, we show that both the model and the gating layer (i.e., the learnable parameters  and ) can converge to stationary points, and the convergence rate is still comparable with existing works. Indeed, the model sparsity  can be further linked to the convergence rate because, intuitively, the larger the  (model size) is, the more local optimal points there are, leading to a faster convergence rate.
>
> We experimentally verified the above intuitive conclusion by showing the convergence curves in Figure 7 in the Appendix part. As shown in Figure 7, the pFedGate with  = 1 converged faster than others, and among the compared pFedGate baselines, it has the largest trainable parameter size. To our best knowledge, the precise relation between the size of the sparse model and the convergence rate still remains to be an open question [1], and we will leave it as future work.
>
> [1] Mean-field theory of two-layers neural networks: dimension-free bounds and kernel limit. COLT, 2019.
>
> **[W4 comparison to FedMask] "This paper must compare of FedMask[1] in terms of masking characteristics by varying sparsity."**
>
> Thank you for your comments! Following your suggestion, in addition to the comparison of experimental results in Appendix H.4, we now provide a more detailed discussion of the differences between the two methods as follows:
>
> Note that FedMask freezes the local models during the whole FL process and only learns the binary masks for the last several layers of the models. Instead, we train the both the local models and the personalized gating layers, which generate continuous gated weights to mask some sub-blocks and scale the weights of other sub-blocks, leading to high capacities for the learnable personalized models.
>
> With a smaller sparsity hyper-parameter value, the size of model's hypothesis space searched by FedMask (depicted by only the masks for the last several layers) is much smaller than the one of pFedGate (depicted by both the whole model and the gating weights for all the sub-blocks). Empirical evidence by varying sparsity is shown in Table 7, where we can see that with the same sparsity, pFedGate gains much better accuracy than FedMask (e.g., 86.75 v.s. 77.41 on FEMNIST with ). Besides, when increasing the sparsity ( to ), pFedGate still achieves good or even better performance (the accuracy change is +0.248 on average) while FedMask gains large performance drops (the accuracy change is -1.543 on average).
>
> **[Dense presentation] “… appears quite dense, so I'm concerned about violation of the ICLR2023 template"**
>
> Thank you for your comment! Indeed, this paper has already conveyed a lot of content (20+ pages) and our previous layout made the main page a little dense. Following your suggestion, we have now made some adjustments to simplify the related work section and put more detailed comparisons in the Appendix to make the main page fit the template.

---

> ### Author Response · Authors · 2022-11-28
> **Responses to Reviewer eJXn, part 3/5.**
>
> **[W2.3] "how extents of sparsity this model can support."**
>
> 3. [**"boundary of this method by changing communication frequency and data distribution"**]
>     - According to your suggestion, we newly conduct the experiments by (1) changing the local update steps (the smaller the number of steps, the more frequently the Federation updates) and (2) changing the Dirichlet factor $\alpha$ to simulate different Non-IID degrees (the smaller the $\alpha$, the higher the heterogeneity of the data). The results on EMNIST with different local update steps and results on CIFAR-10 with different $\alpha$s are listed in the below two tables respectively.
>     - Generally speaking, we can find that pFedGate gains comparable performance when the local update step increases from 1 to 4, but a significant drop when the step becomes 8. Besides, as the Non-IID degree increases, the accuracy of pFedGate mostly decreases especially when $\alpha=0.01$. These observations suggest that there is still room to improve the robustness of pFedGate, particularly in terms of how to effectively aggregate the local models for gaining high accuracy when the local and global models can easily differ greatly. For example, in the future, we could consider combining with asynchronous federal learning [1] and more advanced aggregation mechanisms in the model parameter space [2] to allow for larger variations of the participants' models.
>
> | FEMNIST | Step=1  |            | Step=2  |            | Step=4  |            | Step=8  |            |
> |---------|---------|------------|---------|------------|---------|------------|---------|------------|
> |         | Acc-avg | Acc-bottom | Acc-avg | Acc-bottom | Acc-avg | Acc-bottom | Acc-avg | Acc-bottom |
> | s=1     | 87.11   | 81.43      | 86.87   | 81.21      | 87.01   | 81.07      | 84.28   | 79.53      |
> | s=0.5   | 87.28   | 81.15      | 87.25   | 80.99      | 87.09   | 80.65      | 84.62   | 78.45      |
> | s=0.3   | 87.09   | 82.52      | 87.48   | 82.63      | 86.39   | 82.08      | 85.31   | 80.86      |
> | s=0.1   | 83.58   | 75.14      | 83.27   | 74.81      | 82.99   | 74.58      | 80.75   | 73.05      |
>
> | CIFAR-10 | a=0.4   |            | a=0.2   |            | a=0.1   |            | a=0.01  |            |
> |----------|---------|------------|---------|------------|---------|------------|---------|------------|
> |          | Acc-avg | Acc-bottom | Acc-avg | Acc-bottom | Acc-avg | Acc-bottom | Acc-avg | Acc-bottom |
> | s=1      | 75.18   | 66.67      | 74.29   | 65.71      | 73.55   | 65.23      | 72.63   | 64.64      |
> | s=0.5    | 74.07   | 64.21      | 73.05   | 63.44      | 72.55   | 63.03      | 71.76   | 62.02      |
> | s=0.3    | 73.65   | 64.39      | 72.58   | 64.05      | 71.95   | 62.83      | 71.58   | 62.68      |
> | s=0.1    | 70.26   | 60.45      | 69.74   | 59.56      | 68.96   | 59.38      | 67.45   | 58.04      |
>
> [1] Federated Learning with Buffered Asynchronous Aggregation. AISTATS 2022.
>
> [2] Git Re-Basin: Merging Models modulo Permutation Symmetries. ArXiv 2022.
>
>
>
> **[W3.1 sparsity in theorems 1.] "in theorem 1 and 2, they do not consider the sparsity of each client, as well as not provide any high-level understanding of this sparsity based on those theorems"**
>
> Many thanks for your insightful comments! For theorem 1, in the response stage, we revisited its derivation carefully and succeeded in introducing the sparsity factor $s_i$ into it. Specifically, in the original derivation of the inequality in lines 7 to 8 of Eq. 15, we eliminated the mask $M_i$ term on the right-hand side because it has a maximum value of 1 ($\phi_i$ applies sigmoid activation in the final outputs). But actually, considering the sparsity of $\theta'_i$ in the derivation as $s_i$, here the `1` can be replaced by a tighter, more informative value $s_i$.
>
> By following the same idea of the remaining steps of the original derivation, and replacing the $1$ of the rightmost term into $s_i$, we can obtain a new version of the results of Theorem 1, which is: "there exist $\tilde{N}=\mathcal{O}\left( \frac{d}{|\mathcal{C}|\epsilon^2}\log \frac{RL_fL_h(RL_{\textit{g}} +s_i)}{\epsilon}  + \frac{d_{\phi}}{\epsilon^2}\log\frac{RL_fL_h(RL_{\textit{g}} +s_i)}{\epsilon}  - \frac{\log \delta}{|\mathcal{C}|\epsilon^2} \right)$, ...."
>
> Note that in addition to the previous discussion of Theorem 1 still holding, we can currently observe that $s_i$ appears in the numerator of the two log terms. From the new results, we can see that **as the sparsity $s_i$ decreases, the smaller the $\tilde{N}$, and the better the generalization**. This is also to some extent a reflection of Ockham's Razor in our approach. There is similar corroboration in other scenarios, such as the positive effect of the sparsity regularity on generalization ability [1]. We will add these new results and discussion in the final version, thanks again!
>
> [1] Structured Sparsity and Generalization. JMLR, 2012.

---

> ### Author Response · Authors · 2022-11-28
> **Responses to Reviewer eJXn, part 2/5.**
>
> **[W2.1 & W2.2] "how extents of sparsity this model can support.""when a gating function rapidly deteriorates" ... "characteristic of representation capability between global and local models."**
>
> Thank you very much for the insightful comments! We have made the following experiments to explore the extents of sparsity pFedGate can support:
> 1. [**"when a gating function rapidly deteriorates"**]
>     - Actually, in the Ablation study Section (Sec.7.4 and Figure 5), we have examined the extent to which the sparsification still yields acceptable accuracy via reducing $s$ from 1 to 0.1. Generally speaking, as $s$ decreases, the variance of accuracy increases and the accuracy decreases. The accuracy degradation becomes significant when using very small sparsity (from $\bar{ACC}= 87.3$ with $s=0.5$ to $\bar{ACC}=83.6$ with $s=0.1$, which is still acceptable and larger than the one $\bar{ACC}=82.5$ gained by FedAvg).
>     - Recall that we adopt a block-wise sparsification technique and a operator-type-free block splitting manner (Sec.5.2.2), which simply splits each learnable operator of the model into $B$ sub-blocks ($B=5$ by default) and leads to a relatively limited hypothesis space when $s$ is small. To alleviate the performance degradation, in the future, we can explore more fine-grained splitting manner to enhance the expressive ability of the sparse candidate models with very small $s$, such as designing operator-specific strategy for different channels, filters, etc.
>
> 2. [**"characteristic of representation capability between global and local models."**]
>     - According to your suggestion, we newly conduct the experiments that compare the accuracy of both the local (personalized) models and the global (shared) model. We firstly directly examined the accuracy of the global model for the same number of FL rounds when the local model achieved the best performance, and find that the accuracy of the global model can be very bad as in the proposed pFedGate method, the global model does not receive a direct correction signal for the ground truth (but rather via a sparse adaptation). We thus train the global model with one more local epoch, then aggregate the trained one and evaluate it. The results on EMNIST dataset are listed in the Table below.
>     - Interestingly, we found that pFedGate's local model consistently achieved better accuracy than the global model, the accuracy advantage is more significant on Acc-bottom than on Acc-avg. Furthermore,  as the sparsity decreases, the advantage increases (from $s$=1 to 0.3) and then decreases ($s$=0.3 to 0.1). This suggests that introducing moderate sparsity helps generalization, but at very high sparsity our method is at risk of degradation and that the degradation outweighs the benefits of increasing sparsity, requiring further enhancement with small $s$. A discussion of the relevant theoretical intuition will be given later in our response to Theorem 1 ("Responses to Reviewer eJXn, part 3/5").
>
> |                 |      Local Model |        |     Global Model  |      | Local-Global Gap  |      |
> |-----------------|---------|------------|---------|------------|---------|------------|
> |                 | Acc-avg | Acc-bottom | Acc-avg | Acc-bottom |  Gap-Acc-avg | Gap-Acc-bottom |
> | s=1   |  87.11  |    81.43   |  79.47  |    71.28   |   7.64       |    10.15       |
> | s=0.5 |  87.28  |    81.15   |  76.46  |    66.98   |   10.82      |    14.17       |
> | s=0.3 |  87.09  |    82.52   |  77.14  |    66.29   |   9.95       |    16.23       |
> | s=0.1 |  83.58  |    75.14   |  76.45  |    66.98   |   7.13       |    8.16        |

---

### Public Comment · ~Rivan_Ilk1 · 2022-11-19
**To reviewers and authors, concerns regarding the claims and contributions of the paper**

Dear authors and reviewers,
The paper is an interesting read, and the reviewers’ comments are also quite insightful towards understanding the claims of the paper. In this regard, the following points may be helpful for further evaluation of the current work.

- **Limited novelty and potentially ambiguous claims**

**The idea of learning sparse models in a federated setting has been considered in many prior works [1], [2]**. In particular, these works do not require any additional step of model compression after the training procedure is completed. The current work has neither compared with these works nor even cited them anywhere in the paper. Furthermore, these prior works achieve a high accuracy despite having a density of less than 10% (equivalent to s<=0.1 in current paper). The current paper consistently considers much higher levels of density, which makes it unclear if it would perform well at low density. Also the proposed scheme of using a gated mask that adapts to each data point can add significant memory overhead for clients as the entire model has to be present in the memory, both for training and inference.

In continuation with the above comment, **it is not clear how the authors can claim that their proposed approach is able to speed up training or inference, or can save communication bandwidth (the authors have made this seemingly unreasonable claim in the last line of Section 5.1)**.  As the local model density is not that low (~25-30%), the upload communication cost saving is limited. More precisely, due to the irregular nature of the sparsity, there should be additional overhead of indexing causing further reduction of comm. saving.
During downlink, as the global model is dense, there is no comm. benefit. In contrast, [1] leveraged the advantage of sparsity during both up and downlink.

Regarding computation efficiency, it is well-known in pruning literature that when unstructured sparsity is utilized for training/inference models, speedups are not observed unless there is a density of the model is lower than 10% [3]. Since the gated mask is equivalent to unstructured sparsity, **authors’ claims regarding savings in computations during training/inference need to be substantiated through empirical experiments**.

- **Weak theoretical analysis**

**While the attempt of the authors to provide a theoretical analysis is appreciated, the assumptions 1 and 2 are too strong to begin with**. In particular, Assumption 1 does not appear in data distribution form in prior works, which typically make assumptions on the objective functions instead of data distributions. Furthermore, Assumption 2 needs to be further justified. The authors have made the assumption that both the sparse models and the learned gates are Lipshitz bounded, without any justifications. The authors need to provide intuitive reasons behind making such strong assumptions. These assumptions on model parameters/gates do not seem to exist in any prior works, and in absence of any justification of these new assumptions, *the current foundations of theoretical analysis may be considered as weak. This in turn limits the scope of such analysis*.

[1] Bibikar S, Vikalo H, Wang Z, Chen X. Federated dynamic sparse training: Computing less, communicating less, yet learning better. In AAAI, 2022.
[2] Xinchi Qiu, Javier Fernandez-Marques, Pedro PB Gusmao, Yan Gao, Titouan Parcollet, and Nicholas Donald Lane. ZeroFL: Efficient on-device training for federated learning with local sparsity. In ICLR, 2021.
[3] Deep Compression: Compressing Deep Neural Networks with Pruning, Trained Quantization and Huffman Coding. ICLR, 2016 (best paper).

---

> ### Author Response · Authors · 2022-11-28
> **Responses to Rivan, Part 2/2**
>
>
> - "Weak theoretical analysis, "the assumptions 1 and 2 are too strong to begin with" "
>     - Thank you for the comments! For the assumption 1, the assumption 1 (bounded diversity) is necessary to a feasible and effective FL process, in which there is knowledge that can be shared and mutually beneficial between the FL participants. We would like to clarify that we adopt **a weaker form** than the similar assumption that is also widely used in related literature such as [1,2,3,4]. For example, let's considering the assumption 3 in [2]: "(Bounded diversity). The variance of local gradients to global gradient is bounded as $\frac{1}{N}\sum_{i=1}^N ||\nabla f_i(w) - \nabla_\theta f(w)||^2 \leq \sigma^2_f. $ ". By contrast, our assumption is "There exist a $D_{g}$ such that the variance between local and global gradients is bounded as $||\nabla_\theta f(\theta^*_i) - \nabla_\theta f(\theta^*_g)||^2 \leq \sigma^2_i, ~~~\forall i \in \mathcal{C}$". Note that our assumption is actually **a special case of their form** on sparse model, with our $f(\theta^*_i)$ term corresponding to their $f_i(w)$ term, and our $f(\theta^*_g)$ term corresponding to their $f(w)$ term.
>
>     - As for the assumption 2, we would like to give more intuition and discussion to show why it is reasonable. Note that the bounds of the parameters of the global model and the gating layer are dependent on the hypothesis space of the adopted global model and gating layer, which are controllable by users. For example, we can hard clip their model parameters, and introduce bounded activation functions such as sigmoid into the gating layer structure as the proposed method does. A more specific example of this is our implementation, where we use zero initialization, kaiming-uniform initialization and (0, 1)-uniform initialization for the parameters of adopted global model and gating layer, which leads to all these init parameters are bounded by one of 0, 1, or constants calculated by the bounded `fan_in` (usually determined by the data shape). Then during the FL courses, we clip the gradients at each backpropagation step (via `torch.nn.utils.clip_grad_norm_()`), resulting in the bounded parameters and thus the assumption holds.
>
> [1] On the Convergence of Clustered Federated Learning. arXiv, 2022.
>
> [2] Personalized Federated Learning with Moreau Envelopes. NeurIPS, 2020.
>
> [3] Federated Multi-task Learning Under a Mixture of Distributions. NeurIPS, 2021.
>
> [4] Think Locally, Act Globally: Federated Learning with Local and Global Representations. NeurIPS, 2019.

---

> ### Author Response · Authors · 2022-11-28
> **Responses to Rivan, Part 1/2**
>
>
> Many thanks for your interest on our work and bring the mentioned two related works into our attention! However, there are several misunderstandings and we would like to give the following clarifications for your comments:
>
> - "Limited novelty and potentially ambiguous claims"
>     - **[Misunderstanding for the flow of pFedGate]** "In particular, these works do not require any additional step of model compression after the training procedure is completed." Note that our work is an end-to-end approach to achieve the model compression. That is, the model is dynamically transformed into a sparse one on both the training and inference stages, and the process is lightweight based on the introduced small gating layer and block-wise masking.
>
>     - **[Misunderstanding for the additional memory overhead]** "it can add significant memory overhead for clients as the entire model has to be present in the memory, both for training and inference."
>         - The introduced additional memory overhead is very small as the gating layer structure has a simple and small structure, whose shape is about the multiplication of the dimension of the input feature and the number of the total sub-blocks. For example, the size of gating layer $d_{\phi_i}$ is only 6.3% of the original model for the CNN model used on CIFAR-10. *We would like to point out that pFedGate actually has great potential for further storage savings for multi-task applications.* Considering that if we use one dense original model but learn $T$ distinct gating layers for different $T$ downstream tasks on a client, the storage cost will be $d+T*d_{\phi_i}$, which has much smaller of the model storage cost than saving multiple distinct local models.  We will explore to extend the current sample/batch-level adaptation to task-level adaptation in the future.
>
>         - Besides, note that **FL requires training on client-side**, in addition to the model parameters，the memory cost in FL training process also contains the **intermediate tensors in both the forward and the backward propagation phases**, whose size depends on the model architecture, the shape of the inputs, and the specific implementation. Thus once the model-adaption of pFedGate is completed, *the intermediate tensor involved in the forward and backward processes through the sparse model is correspondingly smaller based on block sparsity*, allowing us to get a lower peak training memory than FedAvg. We have tracked the peak memory in Mb for one client and one FL round, and find that the proposed method indeed improved the memory cost, e.g., when $s$=0.3 and $bz=128$, pFedGate uses 66% peak memory compared with FedAvg, meanwhile 5.32% average accuracy and 4.17% absolute improvements (Table 1. Sec.7.2).
>
>     - **[Misunderstanding for the computation efficiency]** " ..., since the gated mask is equivalent to unstructured sparsity, ..." The gated mask has a shape of total number of sub-blocks, and used to mask whole selected sub-blocks. In other word, the proposed method can is based on structured block sparsity.
>
>     - **[Misunderstanding for the communication efficiency]** "..., due to the irregular nature of the sparsity, there should be additional overhead of indexing causing further reduction of comm. saving ..." As clarified above, the sparsity is in block-level. And thus the overhead for indexing is very small in terms of the number of sub-blocks, e.g., a 2 layer CNN with 64 blocks and splitting factor B=5 requires only a small indexing tensor as 320.
>
>     - **[The related two works]**, we clarify that the proposed solution is highly different from the two mentioned solutions (FedDST[1]， ZeroFL[2]), and the following detailed discussion will be added into our final version!
>         - First, from the view of the studied problem, both of these two works focus on non-personalized FL that finds a single global model to perform well on all clients, while we consider personalized case to learning client-wise models.
>         - Second, both of them don't provide any theoretical analyses for generalization or convergence, while pFedGate perform wells with theoretical guarantees in terms of both generalization and convergence.
>         - Third, from the view of methodology, they are based on element-wise unstructured sparsity while our is based on block-wise structured sparsity, which is easier to gain efficiency benefits in extended implementations. Finally, from the experimental results, we can see that FedDST works in sparsity with 0.5 or 0.8 (their Figure 3) while ours works in even $s=0.1$. The another method ZeroFL reported the results of sparsity 0.1 on CIFAR10 and gained 0.42% accuracy improvement compared to vanilla FedAvg (their Table 1, NIID column), while we gained ~3.02% improvement over FedAvg with sparsity 0.1 (our Table 1).

---

### Author Response · Authors · 2022-11-28
**Summary of the responses**

Dear Reviewers,

We sincerely thank you for your time and efforts in reviewing our paper as well as the constructive and insightful feedback! We carefully read and responded to your comments. During this response period, we have conducted additional experiments, discussions, and modifications, which can be summarized as follows:

- **New experiments and comparison to more related works**
    - (eJXn) results to study the effect of sparsity by varying its value, examining the representation capability, and varying the communication frequency and data distribution.
    - (sfrQ) results about the system efficiency.
    - comparison to 9 related works, including FedMN and FedPNAS (gH8r); Per-FedAvg, FedMD, and two bayesian FL methods (wTvc); FedMask (eJXn); FedDST and ZeroFL (Rivan Ilk)

- **Clarity**
    - (PyZd, eJXn) clarification and discussions about the system efficiency.
    - (wTvc, PyZd) explanations for the differentiability of pFedGate.
    - clarification about the experimental details (PyZd) and step-by-step formulation derivation (wTvc).

- **Theoretical results**
    - (eJXn) new results on Theorem 1 by considering the sparsity $s_i$.
    - (sfrQ) correction on the proof of proposition 1, and the definition on page 6.
    -  discussion on the strong convexity (wTvc), and assumption 1 and Theorem 2 (sfrQ).

Thanks again for your detailed and helpful suggestions! We believe the paper has been further greatly improved with the additional clarification, discussion, and experimental results.

We sincerely appreciate that the reviewer consider increasing the rating scores if our responses addressed your comments. Please let us know if you have any additional questions or suggestions. We are happy to engage more and address them!

---

### Decision · Program_Chairs · 2023-01-20

**Decision:**

Reject

**Justification For Why Not Higher Score:**

Please see part 1.

**Justification For Why Not Lower Score:**

N/A

**Metareview: Summary, Strengths And Weaknesses:**

The paper proposes a method, namely pFedGate, for efficient personalized FL by generating different sparse models accounting for both the heterogeneous data distributions and resource constraints. The paper was discussed extensively in a meeting with the reviewers, as I summarize below.

The reviewers agree that incorporating the sparsity in FL is a promising direction and that the paper made interesting contributions.

However,
- The paper needs to be reorganized to reflect the contributions better. In particular, the main strength of the paper seems to be personalization and the claim about resource and communication efficiency should be down-weighted. This is because sparsity is only effective in boosting efficiency when sending parameters from devises to the server (and potentially during training). But, the server still needs to send the full model to the clients. While the paper claims that devises can have *limited resources* in communication and storage, they still need to store the entire model and in every round they still need to download the entire model. This claim should be revised.

- Regarding the theoretical grantees, the bound became significantly worse after the revision, and the reviewers did not find the authors response on bounded variance convincing.

- Regarding the experiments, only comparison to FedAvg is provided and no memory and time comparison is provided to other methods. Hence, it’s not clear how much time is saved in total compared to other methods.